# Incomplete lineage sorting of segmental duplications defines the human chromosome 2 fusion site early during African great ape speciation

## Graphical abstract

## Authors

Zikun Yang, Lu Zhang, Xinrui Jiang, ..., Qiang Sun, Evan E. Eichler, Yafei Mao

## Correspondence

qsun@ion.ac.cn (Q.S.),
ee3@uw.edu (E.E.E.),
yafmao@sjtu.edu.cn (Y.M.)

## In brief

Yang et al. present the structure and evolution and investigate the functional consequences of human chromosome 2 fusion, suggesting incomplete lineage sorting of complex regions during great ape speciation.

## Highlights

- Human chr2 fusion involved segmental duplications, inversions, and repeat turnover

- The fusion event is driven by ILS and occurred ∼5–7 million years ago

- The fusion was formed by SDs and subtelomeric repeats rather than telomeric sequences

- Fusion site depletion alters gene expression in human neural progenitors

Yang et al., 2026, Cell Genomics 6, 101079
January 14, 2026 © 2025 The Author(s). Published by Elsevier Inc.

 CellPress

CellPress

# Incomplete lineage sorting of segmental duplications defines the human chromosome 2 fusion site early during African great ape speciation

Zikun Yang,[1,2,3,18] Lu Zhang,[4,5,6,7,18] Xinrui Jiang,[1,18] Xiangyu Yang,[1,18] Kaiyue Ma,[1] DongAhn Yoo,[8] Yong Lu,[4] Shilong Zhang,[1,3] Jieyi Chen,[1] Yanhong Nie,[4] Xinyan Bian,[4] Junmin Han,[1] Lianting Fu,[1,3] Juan Zhang,[1] Mario Ventura,[9] Guojie Zhang,[3,10,11] Qiang Sun,[4,12,13,*] Evan E. Eichler,[8,14,*] and Yafei Mao[1,3,15,16,17,19,*]

[1]Bio-X Institutes, Key Laboratory for the Genetics of Developmental and Neuropsychiatric Disorders, Ministry of Education, Shanghai Jiao Tong University, Shanghai, China
[2]Zhiyuan College, Shanghai Jiao Tong University, Shanghai, China
[3]Center for Genomic Research, International Institutes of Medicine, Fourth Affiliated Hospital, Zhejiang University, Yiwu, Zhejiang, China
[4]Institute of Neuroscience, Center for Excellence in Brain Science & Intelligence Technology, Chinese Academy of Sciences, Shanghai, China
[5]School of Life Science and Technology, ShanghaiTech University, Shanghai, China
[6]Lingang Laboratory, Shanghai 200031, China
[7]Shanghai Center for Brain Science and Brain Inspired Intelligence Technology, Shanghai, China
[8]Department of Genome Sciences, University of Washington School of Medicine, Seattle, WA, USA
[9]Department of Biosciences, Biotechnology and Environment, University of Bari Aldo Moro, Bari, Italy
[10]Center of Evolutionary & Organismal Biology, and Women's Hospital at Zhejiang University School of Medicine, Zhejiang University, Hangzhou, Zhejiang, China
[11]University School of Medicine, Zhejiang University, Hangzhou, Zhejiang, China
[12]Key Laboratory of Genetic Evolution & Animal Models, Chinese Academy of Sciences, Kunming, China
[13]University of Chinese Academy of Sciences, Beijing, China
[14]Howard Hughes Medical Institute, University of Washington, Seattle, WA, USA
[15]Shanghai Jiao Tong University Chongqing Research Institute, Chongqing, China
[16]Shanghai Key Laboratory of Embryo Original Diseases, International Peace Maternity and Child Health Hospital, School of Medicine, Shanghai Jiao Tong University, Shanghai 200030, China
[17]Center for Comparative Biomedicine, Ministry of Education Key Laboratory of Systems Biomedicine, State Key Laboratory of Medical Genomics, Institute of Systems Biomedicine, Shanghai Jiao Tong University, Shanghai, China
[18]These authors contributed equally
[19]Lead contact
*Correspondence: qsun@ion.ac.cn (Q.S.), ee3@uw.edu (E.E.E.), yafmao@sjtu.edu.cn (Y.M.)

## SUMMARY

All great apes differ karyotypically from humans due to the fusion of chromosomes 2a and 2b, resulting in human chromosome 2. Here, we show that the fusion was associated with multiple pericentric inversions, segmental duplications (SDs), and the turnover of subterminal repetitive DNA. We characterized the fusion site at the single-base-pair resolution and identified three distinct SDs that originated more than 5 million years ago. These three distinct SDs were differentially distributed among African great apes as a result of incomplete lineage sorting (ILS) and lineage-specific duplication. One of these SDs shares homology to a hypomethylated SD spacer sequence present in the subterminal heterochromatin of *Pan* but is completely absent subtelomerically in both humans and orangutans. CRISPR-Cas9-mediated depletion of the fusion site in human neural progenitor cells alters the expression of genes, indicating a potential regulatory consequence to this human-specific karyotypic change. Overall, this study offers insights into how complex regions subject to ILS may contribute to speciation.

## INTRODUCTION

Karyotype evolution is a critical aspect of evolutionary biology because it has been associated with speciation, adaptation, and disease.[1–7] In 1935, Painter and Stone first observed that chromosome fusion could lead to speciation in flies.[5] As the field advanced, numerous instances of chromosome fusion have been documented in plant and animal speciation.[1,3,4,6,8–10]

Several molecular mechanisms have been proposed as key drivers of chromosome fusion in speciation and oncogenesis, including telomere-telomere, telomere-centromere, tandem repeat, segmental duplication (SD), and retrotransposon expansions and fusions.[11–15]

Recent advances in genome editing and synthetic biology have provided deeper insights into the role of chromosome fusion in evolution.[16–18] Boeke and colleagues, for example,

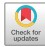

demonstrated that chromosome fusion could induce reproductive isolation in yeast, highlighting the potential of these genomic changes to influence speciation.[19,20] Additionally, studies on chromosome engineering in mice reveal significant chromatin conformation alterations in the chromosome fusion regions, further emphasizing the impact of karyotype evolution on genetic and phenotypic diversity.[21,22]

Human chromosome 2 (chr2) was formed from the fusion of nonhuman primate (NHP) chr2a and chr2b,[23–25] representing arguably the most significant karyotypic difference between humans and NHPs. Ijdo et al. were the first to identify the fusion site at human chromosome 2q13-2q14 using cytogenetic banding approaches.[24] Subsequent studies uncovered dispersed SDs associated with this fusion site.[26,27] These analyses relied on partial genome assemblies generated using short-read technologies or Sanger sequencing of bacterial artificial chromosome (BAC) clones, often resulting in incomplete genetic information at the fusion site.[26–29] Consequently, the structure, evolutionary history, and function of the fusion site have remained only partially understood.

Available genomic assemblies and fluorescence *in situ* hybridization (FISH) experiments have shown an abundance of satellite sequences (pCht sequences) in the subtelomeric repetitive regions of NHP chr2a and chr2b, whereas such sequences are absent in humans.[27,30,31] During the fusion of human chr2, one of the centromeres in the fused chromosome becomes inactive and degraded, with independent transposable element (TE) retrotransposition events occurring at the degenerate site.[32] In addition, there has been considerable debate regarding the timing of the chr2 fusion event. Based on SD and SVA (SINE-VNTR-Alu; SINE, short interspersed nuclear element; VNTR, variable number tandem repeat) divergence, the fusion event was estimated to have occurred early in human evolution, 5–7 and 2.5–4.5 million years ago (mya), respectively.[27,33] However, a recent study, utilizing clustered substitution statistics, proposed a much more recent origin of approximately 0.9 mya.[34]

The complex repetitive structure at the fusion site, subtelomeric repetitive regions, and the inactive centromere have made it challenging to reconstruct the evolutionary history and potential functional consequences of the fusion event. In the absence of complete genome assemblies, most SDs and satellite arrays remain unresolved,[35,36] limiting our ability to examine their evolutionary dynamics. Previous studies using long-read sequencing have shown that SDs can undergo rapid lineage-specific expansion,[37–39] yet no comprehensive analysis has been performed at the human chr2 fusion site. Meanwhile, pCht satellite sequences have been fully characterized in a recent ape genome study, but their detailed relationship with the fusion event remains unclear. Here, we leverage the complete sequences of great apes[38,40] and a macaque[39] (as an outgroup of the great apes) to revisit the structure and evolutionary history of human chr2. We aim to (1) characterize the fusion site at single-base-pair resolution, (2) study this in the context of epigenetic and structural changes occurring at the subtelomeric repetitive regions and degenerate centromere site, (3) use these data to create a model for human chr2 evolution, and (4) examine the potential functional consequences by creating fusion-site-depletion cell lines.

## RESULTS

### Comparative sequence analysis of the human chr2 fusion site

To characterize the syntenic regions and structural changes associated with the fusion site in primates, we performed a comparative analysis of 10 finished nonhuman great ape chromosomes[38] (including chimpanzee, bonobo, gorilla, Sumatran orangutan, and Bornean orangutan) and the finished macaque genome[39] to human chr2 (Figure 1A; STAR Methods). We identified numerous non-syntenic segments (76–154 regions ≥10 kbp in length) between humans and NHPs, accounting for 9.86 Mbp (macaque) and up to 57.5 Mbp (gorilla) of unalignable chromosomal sequence per species. Most of this sequence corresponded to various classes of repetitive DNA (Figure 1A; Table S1), including satellite repeats and tandem and interspersed SDs. In particular, among the nonhuman African great apes, both chr2a and chr2b are acrocentric, with the presence of subterminal heterochromatic caps (5–19.6 Mbp) demarcating the ends of the chromosomes, and are composed of 32 bp AT-rich satellite DNA (pCht) and SD spacer regions.[27,31,38,41] In addition, there are multiple pericentric and paracentric evolutionary inversions distinguishing the ape lineages from each other,[42] several of which share homology with the SD flanking regions of the fusion site (Figures 1B and 2A). One pericentric inversion occurred in the common ancestor of African great apes after divergence from orangutans (chr2b), while another occurred after the *Pan*-gorilla split (chr2a)[42,43] (Figure 1A). Furthermore, we identified the truncated gene *CBWD2*, caused by the truncated SD at the flanking of the chr2a pericentric inversion breakpoint, while a truncated SD but complete gene *FOXD4L1* was found near the chr2b pericentric inversion breakpoint (Figure S1).[28,29] Furthermore, comparative analysis reveals that the pericentric inversions in nonhuman great apes have shaped the gene architecture near the ancestral telomeres. Specifically, the chr2a pericentric inversion disrupted *CBWD2*, resulting in a truncated gene in the *Pan* lineage, whereas the chr2b inversion left *FOXD4L1* intact (Figure S1). These ancestral gene configurations are preserved in humans and flank the chr2 fusion site.

### Evolutionary reconstruction of the ancestral fusion site

To precisely identify the fusion site at single-base-pair resolution and decompose its substructure, we further compared human chr2 with *Pan* chr2a and chr2b to characterize the ∼109 kbp fusion site in the human telomere-to-telomere (T2T)-CHM13v2.0 genome assembly (2q14.1, chr2:113940058–114049496) (Figures 1B and S2). Leveraging data from 47 human genomes from the Human Pangenome Reference Consortium (HPRC),[44] we confirmed that there is a complete absence of large structural variants within this fusion site in 94 sequence-resolved human haplotypes, suggesting a fixed structural haplotype for the entire fusion site (Figure S3). Neither nucleotide diversity (π) nor Tajima's D values show significant reductions at the fusion site in African or non-African populations, relative to the chr2 average (Figure S4), consistent with the locus evolving neutrally.

To gain deeper insight into the evolutionary history of the ∼109 kbp fusion site, we separated this region into three distinct,

CellPress

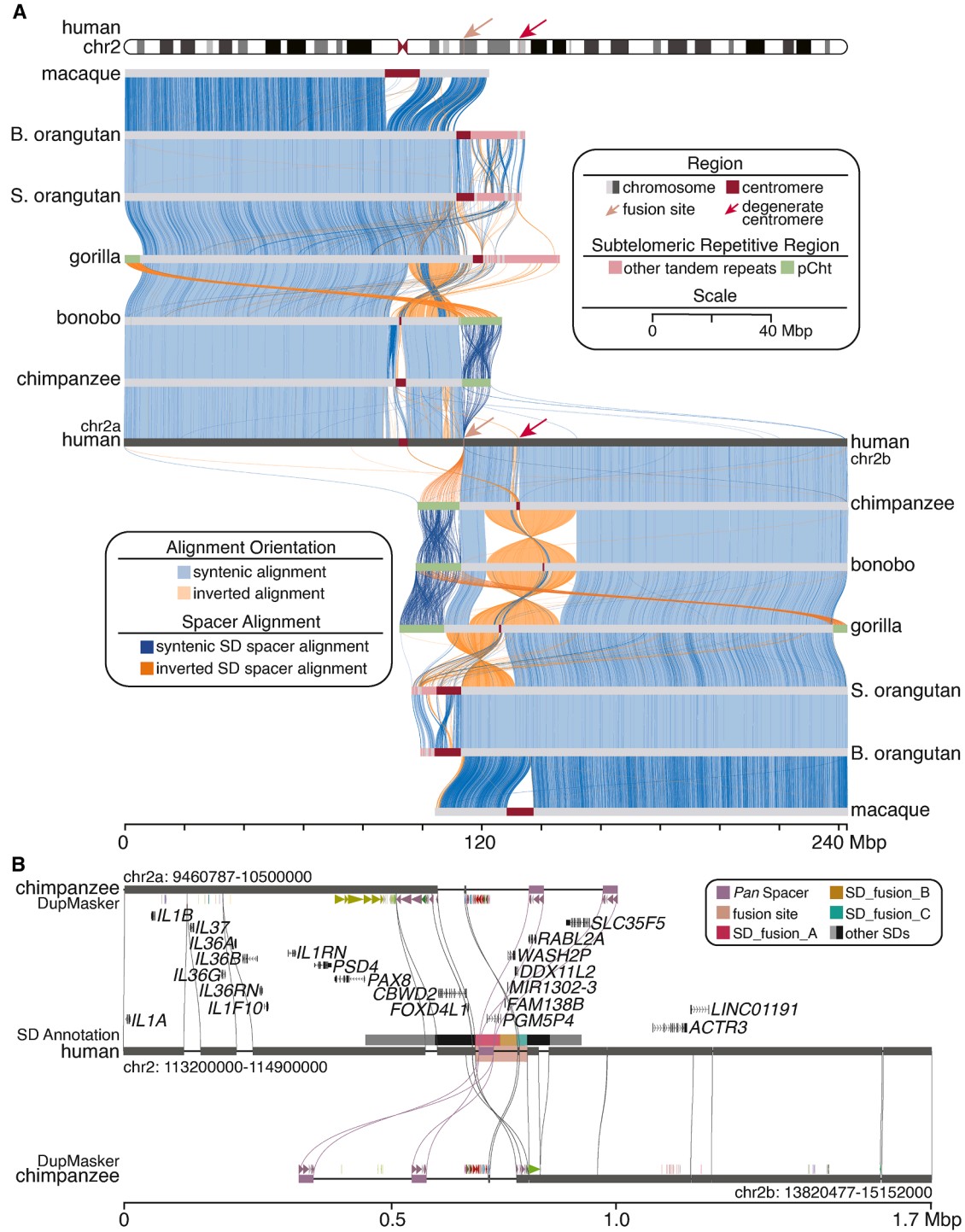

**Figure 1. The comparative sequencing analysis of primate chromosome 2**

(A) The syntenic comparison of chromosome 2 (chr2) highlights the extent of primate chr2 evolutionary rearrangements. Syntenic regions conserved in order (blue) are contrasted with evolutionary inversions (orange) and non-syntenic regions (breaks) corresponding to α-satellite (dark brown), pCht subterminal satellite (green), and other satellite (pink) DNA. B. orangutan and S. orangutan represent the Bornean orangutan and the Sumatran orangutan, respectively.

(B) High-resolution analysis of the human fusion site (chr2:113940058–114049496, colored in amber) shows the non-syntenic breakpoint region in the context of annotated human protein-coding genes and segmental duplications (SDs). *Pan* SD spacers (purple blocks) in chimpanzee pCht show the homology of the partial region of the human fusion site.

See also Figures S1 and S2 and Table S1.

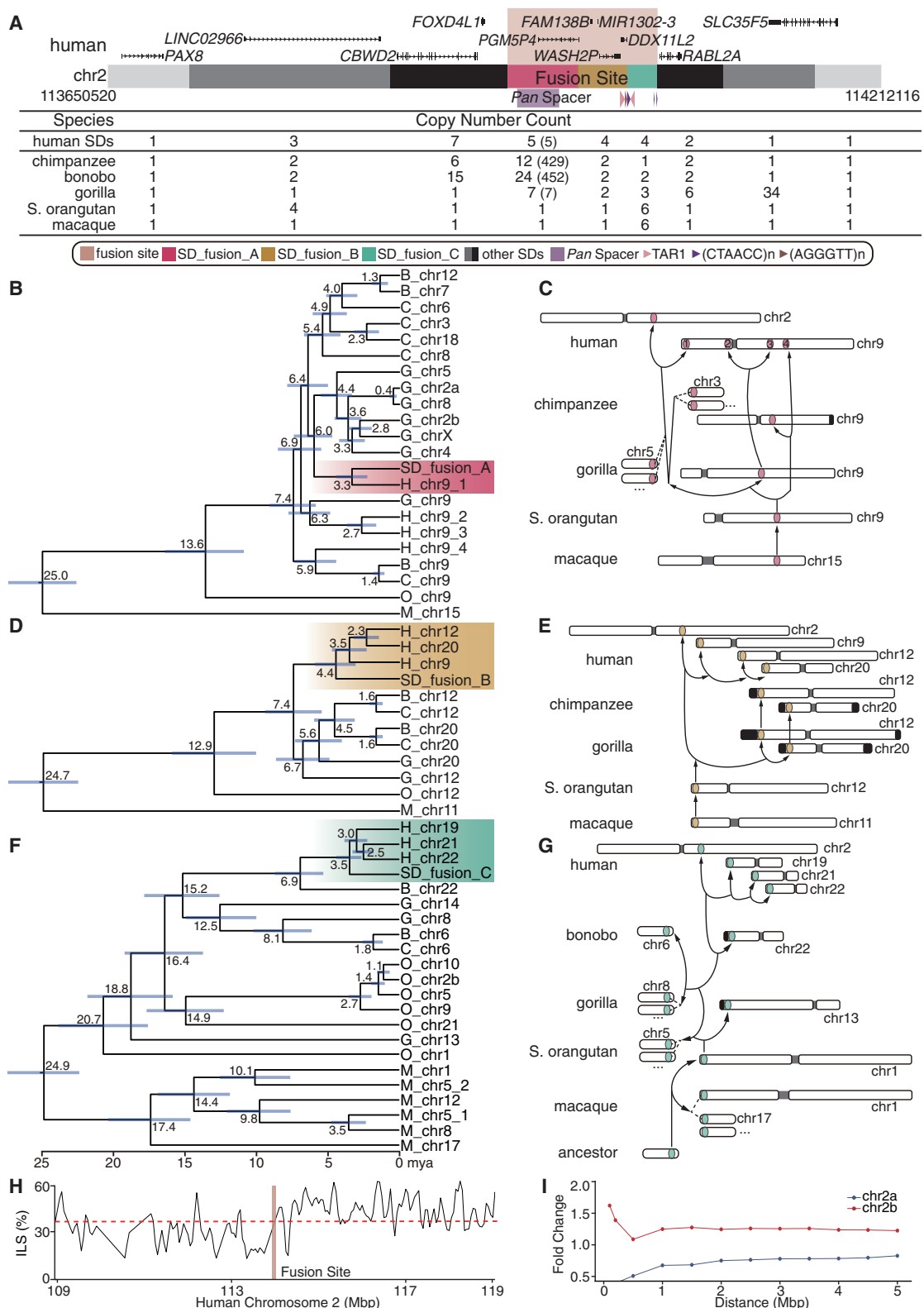

**Figure 2. The genomic structure and evolutionary history of SDs at the human fusion site**

(A) Human chromosome 2 (chr2) fusion site and SD organization. A human genomic segment (chr2:113650520–114212116) with gene annotations shows the region where chr2a and chr2b fused (chr2:113940058–114049496, amber). The region consists of a large (~455 kbp) duplication block made up of seven SDs

*(legend continued on next page)*

high-identity SDs (≥98% identity and ≥20 kbp in length) within the larger ~455 kbp duplication block (seven independent SD blocks) (Figures 1B, 2A, and S5–12; Tables S2A and S2B). In humans, these SDs share homology with sequences corresponding to chr9 (SD_fusion_A, 50 kbp), chr12/20 (SD_fusion_B, 36 kbp), and chr22 (SD_fusion_C, 22 kbp) (Figure 2A; Table S2A). Among African apes, each SD is highly variable in copy number, showing evidence of shared ancestral locations as well as lineage-specific duplications (Table S2A). Using macaque and orangutan as outgroups, we reconstructed the evolutionary history of each SD, estimating the divergence time point from each SD to its nearest nonhuman great ape neighbor (Figures 2B–2G; STAR Methods).

SD_fusion_A (50 kbp) is present as a single copy in macaques and orangutans and corresponds to a partial duplication of *PGM5*[29] (14 exons of an ancestral gene), a gene involved in carbohydrate metabolism, originating from African great ape ancestral chr9 (phylogenetic group IX) (Figures 2B and 2C). This SD began to duplicate in an interspersed configuration in the common ancestral lineage of African great apes (~7.4 mya), with the locus expanding to the highest copy number in chimpanzees and bonobos, where it defines the SD spacer region demarcating large blocks of pCht on chr2a and chr2b and other chromosomes (chr3, chr5–chr13, chr15–chr22, and chrX in chimpanzee; each chromosome in the bonobo genome) (Figures 2B, 2C, and S6). Of note, the SD_fusion_A (chr2) and other human copies share a monophyletic origin (~3.3 mya), most closely related to gorilla copies ~6 mya (95% confidence interval [CI]: 4.8–7.4 mya) instead of chimpanzee or bonobo—a pattern consistent with incomplete lineage sorting (ILS).

Similarly, SD_fusion_B is a 36 kbp segment corresponding to the partially truncated *WASH2*[45] (11 exons of an ancestral gene that potentially regulates actin cytoskeleton dynamics), *FAM138B* (3 exons of an ancestral gene of unknown function), and *DDX11* (3 exons of an ancestral gene implicated in DNA metabolism). The segment exists as a single copy in both macaques and orangutans and originated from a locus mapping to ancestral African great ape chr12 (phylogenetic group XII) (Figures 2D, 2E, and S7). All human copies mapping to human chr2, chr20, chr12, and chr9 show a monophyletic origin (~4.4 mya, 95% CI: 3.1–5.8 mya), suggesting human-specific duplications or interlocus gene conversion events.[46] With respect to nonhuman African great apes, however, this clade diverges from a distinct monophyletic clade that includes chimpanzee, gorilla, and bonobo (~7.4 mya, 95% CI: 5.4–9.3 mya) (Figure 2D). This topology is once again consistent with ILS.

In contrast to SD_fusion_A and SD_fusion_B, SD_fusion_C (the most distal, 22 kbp) shows evidence of independent duplication in all primates, including macaques and orangutans, with the exception of chimpanzees, where it exists as a single copy on chimpanzee chr6 (Figure S8). All human SDs show a monophyletic origin (~3.5 mya, 95% CI: 2.7–4.3 mya) but show a deep coalescence with another genomic segment present only on bonobo chr22 (phylogenetic group XXII), dating back to ~6.9 mya (95% CI: 5.3–8.6 mya) (Figures 2F and 2G). Notably, SD_fusion_C and its flanking region correspond to a larger SD that aligns exclusively between the human chr2 SD_fusion_C site and bonobo chr22 but not with any other NHPs (Figure S13). Thus, SD_fusion_C and its flanking region likely represent an ancestral genomic structure in the latest common ancestor (LCA) of African great apes, which was subsequently sorted in the human and bonobo lineages.

ILS refers to the random segregation of ancestral alleles into descendant lineages, a process that typically arises when the ancestral population size is large.[47–49] ILS results in gene tree discordance and deep coalescence. Previous studies have speculated that ILS may be involved in evolutionary divergence or speciation, although direct evidence and the underlying mechanisms remain to be fully explored.[50,51] Next, we investigated whether ILS also occurs in the regions flanking the fusion site. To this end, we extended the ILS analysis to the 5 Mbp mapping proximally (chr2a) and distally (chr2b) to the fusion site in humans (STAR Methods) using a 500 bp windowed approach[37] (Table S2C). We observe a sharp transition in the proportion of ILS windows at the site of the chr2 fusion. Specifically, we find that the proportion of ILS rises to 50.69% (a 1.39-fold excess compared to the chr2 average) as we approach the fusion site distally (chr2b side). In contrast, the proximal portion appears depleted for ILS segments (Figures 2H, 2I, and S14; Table S2D). There is, thus, a polarized pattern of ILS with maxima and minima occurring on either side of the fusion site.

## The refined analysis of telomeric sequences at the fusion site

Previous investigations identified inverted telomeric sequences (TTAGGG/CCCTAA) at the fusion site, suggesting a T2T fusion.[24] However, whether these sequences are remnants of the ancestral telomeres from chr2 fusion or simply telomeric-like repeats interspersed within ancestral non-telomeric regions remains unresolved. First, we confirmed the presence of interstitial telomeric sequences (CCCTAA, chr2:114027659–114028207) at SD_fusion_C and its corresponding SD on human chr22 (TTAGGG, chr22:51254086–51323279) and similar

---

with variable copy numbers in each ape genome. The three SDs at the fusion site are designated as SD_fusion_A, SD_fusion_B, and SD_fusion_C. The table indicates the copy number of the full-length homologous segments for each primate genome (with brackets indicating the SD_fusion_A copy number of the derived sequence in the subterminal heterochromatic caps of chimpanzees and bonobo; see Table S2A for the detailed alignments at the fusion site).

(B, D, and F) Phylogenetic trees based on a (B) 50 kbp (SD_fusion_A), (D) 36 kbp (SD_fusion_B), and (F) 22 kbp (SD_fusion_C) multiple sequence alignment show that these regions have been subject to incomplete lineage sorting (ILS).

(C, E, and G) The chromosome schematic depicts the genomic locations of SD_fusion_A (C), SD_fusion_B (E), and SD_fusion_C (G) in each primate.

(H) Proportion of human-gorilla and *Pan*-gorilla tree topology in a 500 bp window of 5 Mbp flanking region near the fusion site. The red dotted line represents the mean value of the whole chr2.

(I) Fold change of the mean proportion of discordant topologies in the flanking region compared with the whole genome average. (Note that chromosome names in the great apes refer to the human homologous chromosome, also known as the phylogenetic group designation.)

See also Figures S3–S17 and Table S2.

sequences in bonobo chr22 (TGAGGG, chr22:60722979–60724048). The sequence GGGTTA was identified at SD_fusion_B (chr2: 114027333–114027658) and its corresponding SD on human chr12 (TAACCC, chr12:2843–3030) and bonobo chr20 (TAACCC, chr20:1139021–1139266) but not at its corresponding SD on human chr20 (Figures S15 and S16). These observations argue that these telomeric sequences were present on SD_fusion_B and SD_fusion_C prior to the fusion event. In addition, two telomeric-associated repeats (TAR1) flank the telomeric sequences of SD_fusion_B and SD_fusion_C.[26] Given the genomic structure of TAR1 and the telomeric sequences on these SDs (Figures S16 and S17), we propose that, in the ancestral configuration, each SD contained a TAR1 element and telomeric sequences, arranged in an inverted orientation on ancestral chr2a and chr2b. The end-to-end fusion of these two distinct SDs may have facilitated the fusion of ancestral chr2a and chr2b, leading to the vestigial presence of two TAR1 elements and the CTAACC and GGGTTA repeat motifs at the human fusion site.

### African great ape subterminal satellite repeat expansion and human chr2 fusion

Reconstructing the evolutionary history of the fusion event has been challenging due to extensive SD and lineage-specific turnover of the subterminal heterochromatic caps in *Pan* and gorilla[38] (Table S3A). With the exception of the short arm of gorilla chr2a, the corresponding regions in both *Pan* and gorilla are composed of nearly continuous megabase-pair tracts of satellite DNA interspersed with SD spacers.[38] The satellite sequence is made of a tandem 32 bp repeat motif (pCht)[31] punctuated on average every 287 kbp in *Pan* and every 389 kbp in gorilla by an SD spacer (Figures 3A and S18–S20). The interrupting SD spacers are variable with a modal length of 32 kbp in the *Pan* lineage and 33.7 kbp in gorilla (Figure S18) and correspond to hypomethylated pockets flanked by the hypermethylated satellite DNA[38] (Figures 3A, S19, and S20). Although both the gorilla and chimpanzee SD spacers differ in sequence composition and the subterminal heterochromatic cap is largely thought to have evolved independently in both lineages,[27,38] the net effect is that the subterminal portions of both chimpanzee chr2a and chr2b share 23.6 Mbp of high-identity sequence homology involving both the SD spacers and pCht satellite DNA. In the case of the gorilla, the homology is restricted to the pCht satellite DNA for both arms of gorilla chr2b and the q-arm of gorilla chr2a. The organization of the p-arm of gorilla chr2a differs considerably and is much more similar to the organization found in orangutan, which is enriched in HSatIII-like repeat sequences (Figure 3B), which is consistent with a previous report.[52] It is classified as an acrocentric short arm lacking a nuclear organizing region.[38]

Importantly, the SD spacer that expanded in the subterminal heterochromatic caps in chimpanzee and bonobo shares 98.05% identity with the SD_fusion_A segment mapping at the fusion site on human chr2 (Figure 3C). All 429–452 SD spacers within the *Pan* pCht regions, including those at the ends of these chromosomal regions, are monophyletic in origin, estimated to have expanded approximately ~5.5 mya (95% CI: 4.2–6.8 mya). This subterminal SD spacer (32 kbp) in *Pan* is 18 kbp smaller than SD_fusion_A (50 kbp), where the shared LCA

predates the human-*Pan*-gorilla divergence (Figures 2B and 2C). Comparing the phylogenies of the sequence unique to SD_fusion_A and the sequence shared with the subterminal heterochromatic cap SD spacers shows nearly coincident ILS topology (generalized Robinson-Foulds distance = 0.28, $p = 1.6 \times 10^{-4}$). This indicates that the SD spacers within *Pan* heterochromatic caps are derived from the duplicated sequence that gave rise to ancestral SD_fusion_A. Thus, the hyperexpansion of subterminal satellite DNA associated with subterminal heterochromatic caps in *Pan* and the chr2 fusion are linked genetically with two different evolutionary trajectories and karyotypic consequences in human and *Pan* (Figures 3C, 3D, and S21; Table S3B).

### Centromere retention and degeneration in human chr2

An important consequence of the human chr2 fusion is the inactivation of the ancestral chr2b centromere from NHPs in the human lineage. Here, we refer to this site as Cen_decay (short for centromere degeneration; Figure 4A). Chr2a differs from chr2b by the presence of large tracts of HSatII arrays in humans and HSatIII arrays in *Pan* (Figures S22 and S23). Overall, the active human centromere in humans is more similar to that of the gorilla with respect to suprachromosomal family (SF) organization. Both humans and gorillas possess SF2, whereas chimpanzees and bonobos have SF3.

We compared the active chromosome centromere of chimpanzee chr2b with the structure of the vestigial centromere in humans. We identified three distinct α-satellite arrays (chr2:132644386–132685996) in humans[52] with homology to chimpanzee that had been interrupted by various simple repeats (CCTCTC) and retrotransposon elements (SVA and L1PA3) in the human lineage (Figure 4B). All three satellite arrays were derived from divergent monomeric α-satellite regions ancestral to human and chimpanzee rather than from higher-order repeats (HORs). Our analysis of 94 human genome assemblies[42] identifies five distinct structural haplotypes due primarily to length variation of the first two α-satellite arrays (Figures 4C and 4D; Table S4). A sequence comparison using unique *k*-mers from this region shows that ~97.5% (8,246/8,460) and ~96.7% (8,181/8,460) are also identified in Neanderthal and Denisovan genomes, respectively (STAR Methods), confirming that the centromeric degeneration occurred long before the divergence of modern and archaic humans.[32]

We also compared methylation patterns of the NHP chr2a and chr2b centromeres (Figure 4E). The human α-satellite arrays mapping to the degenerate site show significantly lower methylation levels when compared to typical NHP HORs (excluding centromere dip regions [CDRs], $p = 0.02$), with the exception of bonobo chr2b and gorilla chr2b. HORs in bonobo chr2b and gorilla chr2b are significantly hypomethylated when compared to other NHP HORs ($p = 1.92 \times 10^{-6}$), possibly due to the smaller size of these HORs (lengths: 71 kbp for bonobo chr2b and 106 kbp for gorilla chr2b).

### Functional assessment of the fusion site by depletion

As the fusion site is nearly fixed in the human genome, we speculated that it may confer a functional advantage. To explore this possibility, we examined gene models and expression patterns

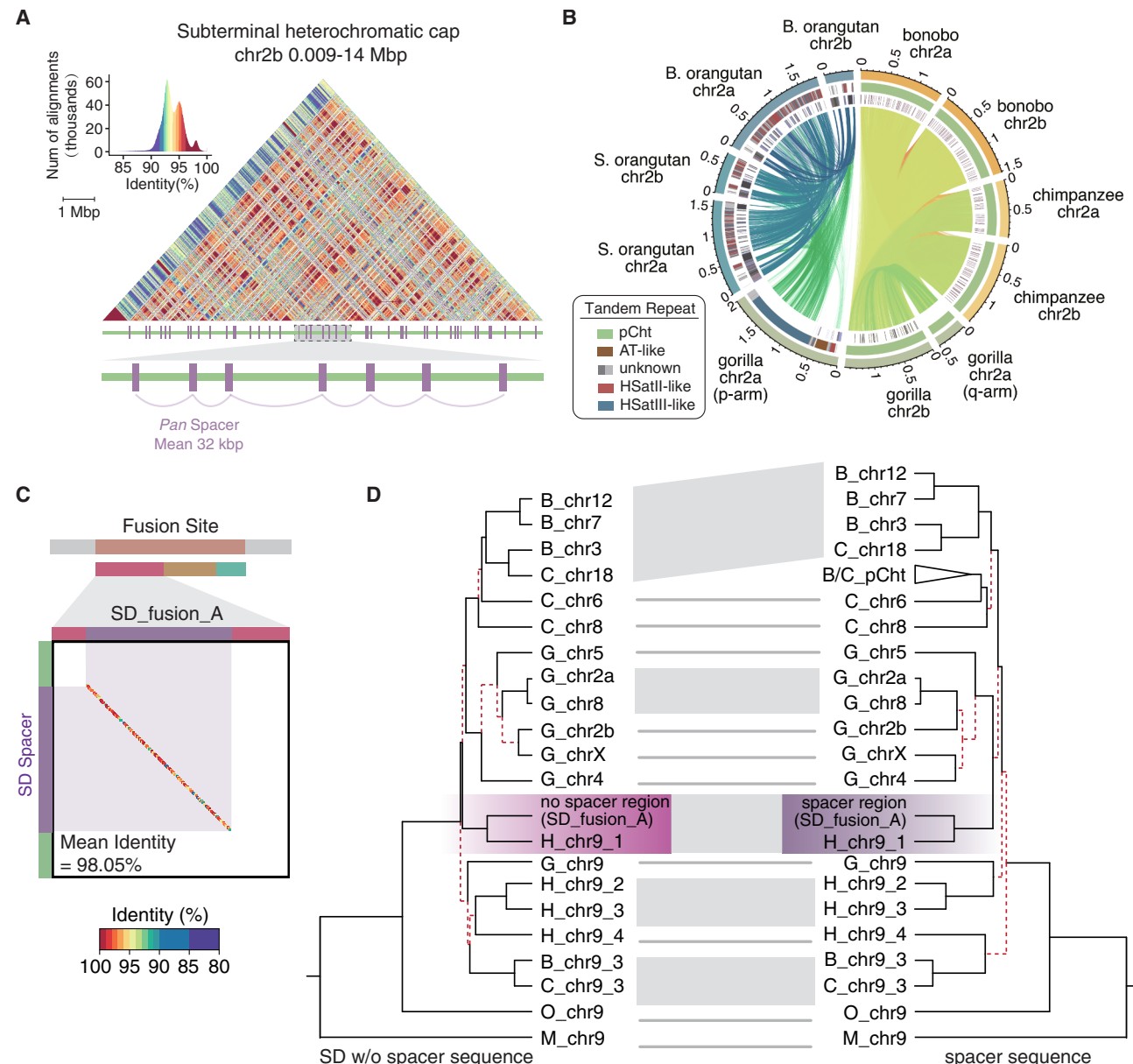

**Figure 3. Turnover of subtelomeric repetitive regions in primates and evolutionary connection between SD spacers in *Pan* lineage and SD_fusion_A at the fusion site**

(A) The identity heatmap of the subterminal heterochromatic cap for the p-arm of chimpanzee chr2b. SD spacer elements (purple) are annotated below and are flanked by large tracts of pCht satellite (green) (zoomed-in image shows the structure at higher resolution).

(B) Circos plot shows sequence identity of chr2a and chr2b among the apes, highlighting the turnover of satellite DNA. The three layers (outer to inner) represent the subtelomeric region, tandem repeat satellites, and transposon element annotations.

(C) Dot plot shows the synteny between a bonobo SD spacer within the heterochromatic cap vs. human SD_fusion_A segment. The sequence identity was calculated at a 100 bp resolution, with the dot color representing sequence identity ranging from 80% (purple) to 100% (red).

(D) Phylogenetic topology comparison shows nearly consistent ILS evolutionary pattern of SD_fusion_A (excluding spacer sequence) and SD spacer. The red lines indicate discordant tree topologies within each major clade between the two trees.

See also Figures S18–S21 and Table S3.

in this region and generated knockout (KO) cell lines to assess potential gene expression changes associated with the fusion site. Four putative noncoding genes/pseudogenes (*PGM5P4*, *FAM138B*, *WASH2P*, and *DDX11L2*) have been annotated at the site of the human chr2 fusion. According to GTEx short-read RNA sequencing (RNA-seq), these four genes/pseudogenes are expressed in testis, esophagus, fallopian tube, and cerebellum tissues[53] (Figure S24). Further, both *PGM5P4* and *WASH2P*

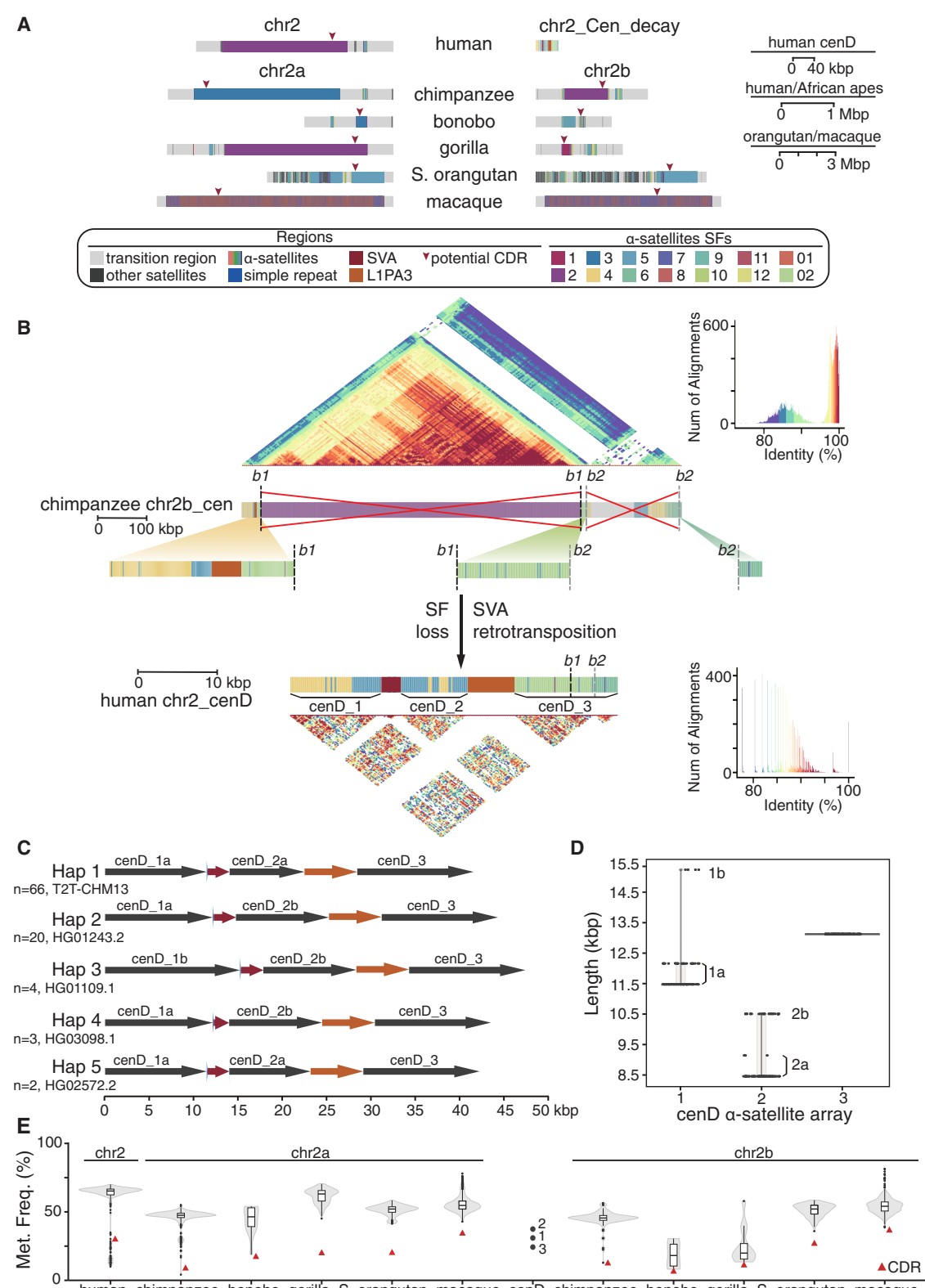

**Figure 4. Comparative analysis of active and inactive centromeric regions of human chromosome 2**

(A) Genomic structure and suprachromosomal family (SF) annotation of centromeres in human chr2, degenerate site (chr2:132644386–132685996), and NHP chr2a/chr2b. Different SFs are shown by various colors, with centromere dip regions (CDRs) marked by arrows.

*(legend continued on next page)*

are supported by long-read isoform sequencing (Iso-Seq) transcript data from CHM13hTERT[54] and kidney tissue from ENCODE.[55] In addition, methylation analysis demarcates a prominent CpG island showing the promoters/enhancers of *PGM5P4*, as identified using ONT reads from the T2T-CHM13 cell line[54] (Figure S25).

To explore the potential function of the fusion site, we used CRISPR-Cas9 to delete this region in CN1[56] induced pluripotent stem cells (iPSCs), followed by directed differentiation into neural progenitor cells (NPCs) for transcriptomic analysis (Figure 5A). One pair of single guide RNAs (sgRNAs; L-sg and R-sg) was designed for the depletion (Figure 5B), and the heterozygous depletion of the fusion site was confirmed by PCR and Sanger sequencing (Figure 5C; Table S5A). For each condition (wild type and depletion), we selected three independent monoclonal cells as biological repeats. We then conducted RNA-seq, generating 100.3, 106.2, and 104.6 million reads for the control cell lines and 137.8, 130.3, and 133.1 million reads for the fusion site deletion cell lines (Figures S26A–S26C; Table S5B).

Differential gene expression analysis identified 547 upregulated and 869 downregulated genes using default settings in the differentially expressed gene (DEG) pipeline (Figures 5D and S26D; Table S5C; STAR Methods). To reduce potential confounding from lowly expressed genes, we excluded DEGs with transcript per million (TPM) values in the lowest 50% of the transcriptome (TPM > 1), resulting in a refined set of 99 upregulated and 178 downregulated genes (Figures S26E and S26H). Functional enrichment analysis revealed that the upregulated genes were significantly associated with pattern specification processes (adjusted $p = 8.61 \times 10^{-5}$) (Figures 5E and S26I), whereas the downregulated genes were enriched for pathways related to neural development and organization, including forebrain development (adjusted $p = 8.02 \times 10^{-15}$) and axonogenesis (adjusted $p = 4.92 \times 10^{-14}$) (Figures 5F and S26J). These findings suggest that deletion of the fusion site may influence transcriptional programs involved in neural development, potentially contributing to phenotypic divergence between humans and NHPs.

## DISCUSSION

In previous NHP assemblies, SD homologous to SD_fusion_A/B/C were largely collapsed (63 out of 74 cases), and subtelomeric repetitive sequences remained unresolved (Figures S27 and S28; Tables S6A–S6C). Complete T2T sequence for chr2, chr2a, and chr2b among the great apes[38–40] (average quality value = 72.1) allowed us to systematically examine the complex genomic architecture and further refine the evolutionary history of the human-specific chr2 fusion event. There are three important conclusions. First, the fusion event was intimately associated with SDs that have been restructuring genomes and chromosomes throughout great ape evolution.[27,28] The 109 kbp fusion site in humans consists of three independent SDs, each with distinct trajectories in different ape lineages, and these are further embedded in a larger duplication block of ~455 kbp. All SDs are highly variable in copy number among the great apes, and many have been reused as breakpoint sequences during great ape evolution. Although cytogenetic studies previously reported two pericentric inversions on ancestral chr2a and chr2b,[27] we observe that the flanking region at the human fusion site (chr2a, proximal side) shares 98.4% identity with an SD flanking the pericentric inversion that distinguishes Sumatran orangutan chr2b from gorilla chr2b. Moreover, SD_fusion_A shares 98.05% identity with the SD spacer sequence of the *Pan* subterminal heterochromatic caps, where it expanded to hundreds of copies in the chimpanzee and bonobo lineages but not in gorillas.

Second, we show that the fusion site has been strongly subjected to ILS. All three SDs, for example, show phylogenetic signatures consistent with ILS with a maximum occurring at the fusion site, suggesting that the fusion event occurred >5 mya. Thus, the fusion event is not a recent evolutionary event but potentially occurred during African great ape speciation when the effective population size was predicted to be much larger than contemporary ape populations.[37,38] Given fossil evidence that *Australopithecus* existed around 2–4 mya, *Paranthropus* around 1–3 mya, and the earliest *Homo* fossils around 2–3 mya,[57–59] we speculate that this fusion event probably did not arise in the genus *Homo* but rather occurred in ancestral great ape populations.

Third, the fusion site was associated with extensive subtelomeric satellite turnover and epigenetic differences among great apes. The two satellite motifs present in the subtelomeric regions of gorilla chr2a and chr2b are distinct from each other: one resembles the subtelomeric repetitive regions of orangutans, while the other resembles the genomic architecture of *Pan* species. SD_fusion_A, in part, defines the chr2 fusion site but also associates with large tracts of hypermethylated pCht satellite repeats that define subterminal heterochromatin in chimpanzees and bonobos[38] (Figure S29). The juxtaposition of novel SDs at the fusion site and the shift from heterochromatic or acrocentric DNA at the termini of chr2a and chr2b led to methylation differences among the great apes at this locus. Indeed, our depletion experiments in humans show consistent gene expression changes, suggesting that the euchromatization of the fusion site may have had more global genome-wide regulatory effects.

It is well established that fusion events can be mediated by SDs and other repetitive sequences.[52,60,61] However, to our knowledge, this study presents the first evidence of SDs subject

(B) Comparison of the centromere in chimpanzee chr2b with the human degenerate centromeric region. The middle image shows the SF annotations for the chimpanzee centromere and human centromere degenerate region. Potential α-satellite deletion breakpoints (*b1* and *b2*) are indicated with dotted lines. The heatmaps of each region are shown on the top and bottom, respectively.

(C) Five distinct structural haplotypes of the degenerate centromere site in humans.

(D) Lengths of three α-satellite arrays in humans.

(E) The methylation degree of HOR across NHPs and of three α-satellite arrays at the degenerate centromere site. The red triangle represents the methylation frequency of CDRs in each chromosome (Met. Freq., methylation frequency).

See also Figures S22 and S23 and Table S4.

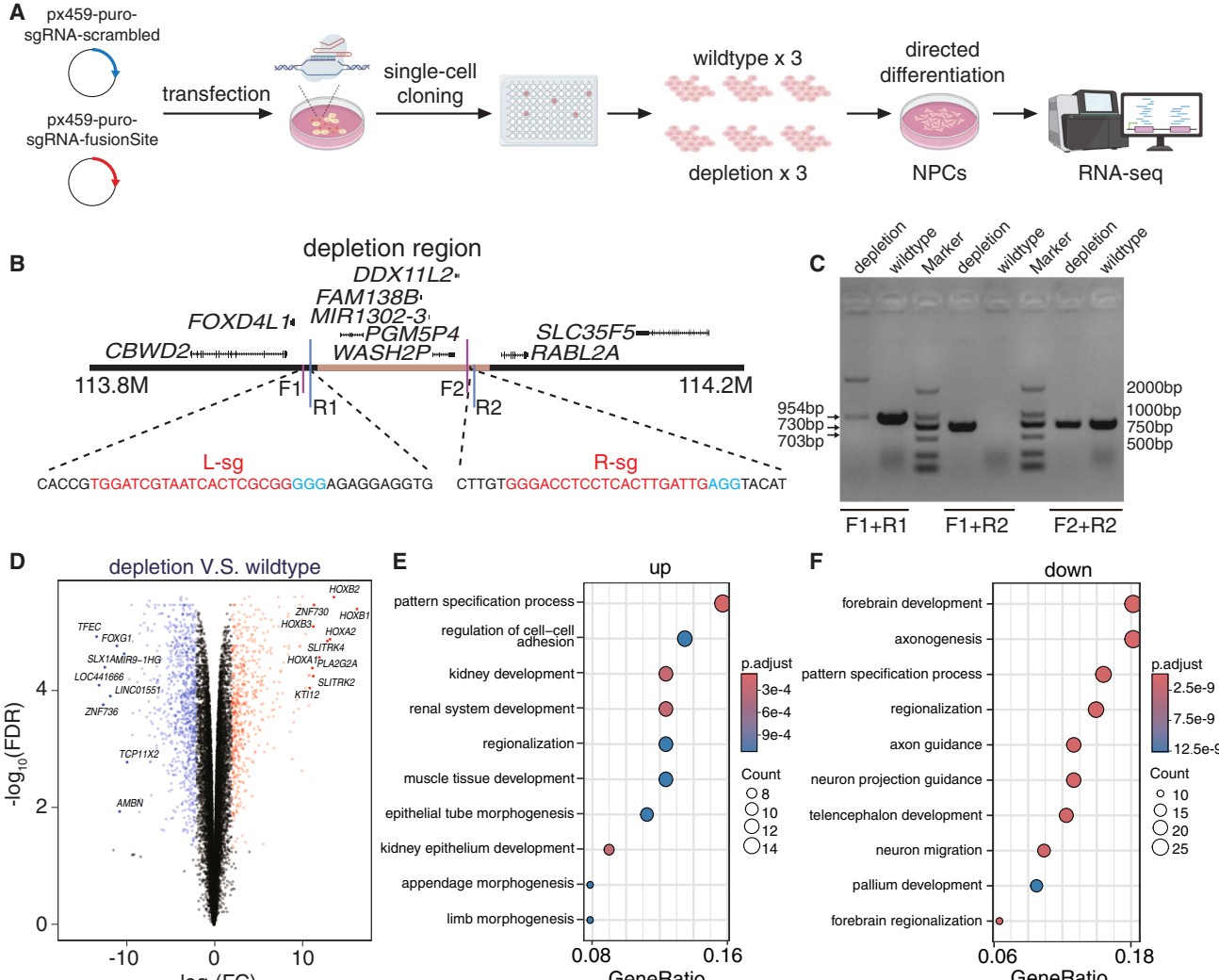

**Figure 5. Fusion site knockout and gene expression alteration**

(A) Schematic representation of the depletion experiments performed in CN1 cell lines. NPCs, neural progenitor cells.

(B) One sgRNA pair (L-sg and R-sg) was designed for the depletion experiments. Red sequences indicate sgRNA, while blue sequences indicate PAM sites.

(C) The PCR analysis confirms heterozygous depletion in cell lines. WT, wild-type cell lines.

(D) Differentially expressed genes (DEGs) are shown in volcano plot. Genes with $\log_2$ fold change ($\log_2$FC) values > 2 and false discovery rate (FDR) < 0.05 are highlighted in red (upregulated) and blue (downregulated).

(E and F) The GO enrichment results of (E) upregulated and (F) downregulated DEGs. Adjusted *p* value is computed and visualized by color from red to blue. See also Figures S24–S26 and Table S5.

to ILS being involved in a speciation or evolutionary divergence event. Based on our observations, we propose two evolutionary scenarios for the formation of human chr2. In the first, the large effective population size of the human-*Pan*-gorilla ancestral population facilitated the coexistence of several NHP chr2a and chr2b subtelomeric structural haplotypes 5–7 mya. In one of these structural configurations, SD_fusion_A and SD_fusion_B became juxtaposed and duplicated to the subtelomeric region of human-*Pan*-gorilla ancestral chr2a prior to the fusion event (Figure 6A). Similarly, SD_fusion_C was duplicated to the subtelomeric region of ancestral chr2b (where the full-length structure is still retained subtelomerically in bonobo); yet, it is no longer

located on other nonhuman great ape chr2b due to the exchange of subtelomeric SDs and evolutionary turnover of satellite DNA.[27,62] Subsequently, the telomeric and TAR1 sequences in SD_fusion_B and SD_fusion_C mediated the fusion (Figure 6A). In the *Pan* and gorilla lineages, chr2a and chr2b (only in chimpanzee and bonobo) experienced a different evolutionary trajectory associated with the formation of subterminal heterochromatic caps. Previous studies[27,38] have shown that the heterochromatic caps likely evolved independently in both gorilla and chimpanzee, albeit convergently with a similar architecture where hundreds of kilobase pairs of satellite pCht DNA are punctuated by a ~30 kbp SD spacer region that defines a pocket of

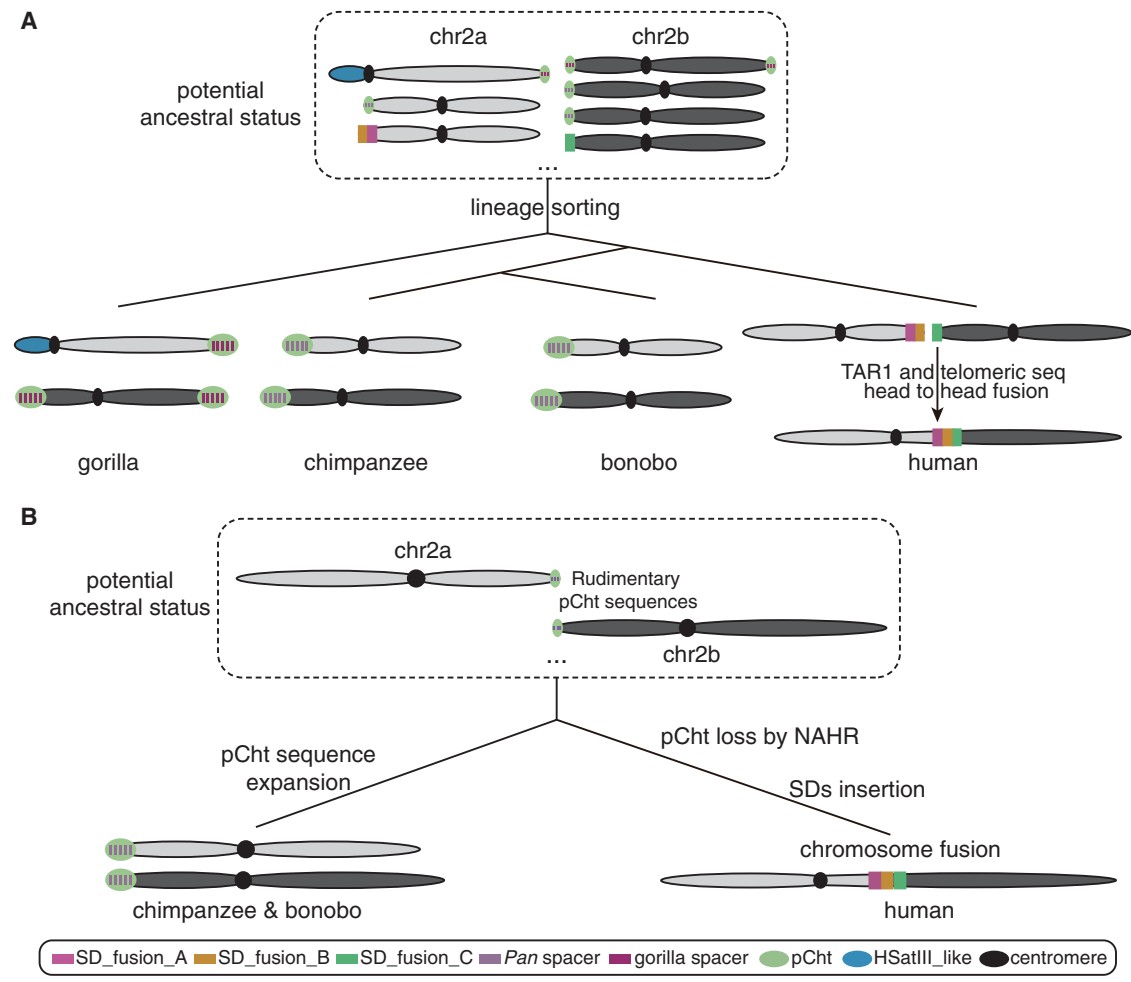

**Figure 6. Models for chromosome 2 evolution**

Two different models are depicted for the origin of the human chromosome 2 (chr2) fusion.

(A) In the last common ancestor of humans, bonobos, chimpanzees, and gorillas, diverse ancestral genomic structures emerged at the ends of chr2a and chr2b. In the human ancestral lineage, chr2a and chr2b consisted of complex SD blocks that emerged as a result of ILS and then became juxtaposed by a telomere-to-telomere fusion.

(B) Rudimentary pCht sequences were present in the ancestral lineage of humans and chimpanzees in association with SD_fusion_A. Nonallelic homologous recombination (NAHR) occurred between these in the human lineage, eliminating pCht in humans but retaining SD_fusion_A, with other SDs subsequently duplicating to the location. In chimpanzees, an independent expansion of the SD spacers and pCht sequences occurred, leading to the formation of the subterminal heterochromatic caps.

See also Figures S27–S30 and Tables S6A–S6C.

hypomethylation. The SD spacers in gorilla and chimpanzee subterminal heterochromatin caps are distinct, but in both chimpanzee and bonobo, the SD spacers represent a derivative of the SD_fusion_A, confirming that this sequence was present subtelomerically for chr2a in the human-*Pan* ancestor.

Alternatively, the SDs evolved similarly, but in the common ancestor of human and *Pan*, there was an incipient association with rudimentary pCht satellite sequences with the SD_fusion_A sequence that mediated the exchange between ancestral chr2a and chr2b via nonallelic homologous recombination (NAHR) or ectopic exchange. Subsequently, during NAHR between full-length SD_fusion_A elements, the pCht regions were lost in the human lineage, and two additional SDs were inserted at the fusion breakpoint region (Figure 6B). In the *Pan* lineage, a portion of the SD_fusion_A became hyperexpanded, defining the subterminal SD spacer region of all heterochromatic caps in bonobo and chimpanzee. In support of this model, it is known that in both chimpanzee and gorilla, the subterminal satellite DNA forms unique post-bouquet structures in germ cells and are hotspots of ectopic exchange between nonhomologous chromosomes.[63] If such exchanges occurred at the edge of the incipient subterminal heterochromatic caps before there were many copies of the subterminal satellite DNA, it would help explain the absence of the pCht sequence in the human genome—i.e., the fusion of chr2a and chr2b helped eliminate the potential for the formation of subterminal heterochromatic caps.

These two models differ in terms of which repetitive sequences played the primary role in mediating the fusion event: the first model favors SDs, while the second emphasizes pCht-like sequences. However, both models support the presence of SDs in the LCA and suggest that ILS, potentially facilitated by a large effective population size, played a key role.[37,38] These aspects helped maintain and diversify karyotypic structures of the ancestral chr2a and chr2b for possibly millions of years. The divergent fates of the subtelomeric repetitive regions of these ancestral chromosomes, e.g., pCht expansion and SD spacer insertions in the *Pan* lineage or the SD insertions in the human fusion site, may have contributed to divergence between humans and NHPs, potentially involving ILS (Figure S30). In this light, it is interesting that we document an asymmetric ILS pattern, with an increase in ILS segments on the distal side (chr2b) and a decrease on the proximal side (chr2a) of the fusion site. This suggests diverse evolutionary trajectories for the ancestral subtelomeric regions of NHP chr2a and chr2b, highlighting the potentially important role of both SDs and ILS in understanding chromosomal evolution, evolutionary divergence, speciation, and sequence turnover.

### Limitations of the study

Speciation is not an instantaneous process—it typically requires extended periods of population divergence and is often more complex than expected. Our study shows that SDs have undergone lineage-specific sorting in extant apes, including humans, and may have played a key role in facilitating the chr2 fusion in the (ancestral) human lineage. However, the exact timing of when this fusion became nearly fixed in the (ancestral) human population remains unknown, but our results suggest that it is ancient and that it occurred early in African ape speciation >5 mya.

Our study supports the idea that ILS plays a crucial role in fusion site formation based on current genomic data from living apes. However, the large population size of the common ancestor of great apes complicates the evolutionary scenario. For example, recent studies have shown that ghost introgression occurred during great ape speciation.[64,65] Therefore, we have sufficient reason to guess that ILS is not the only evolutionary force involved in this fusion formation, and other forces (e.g., ancestral introgression) could also be associated with the fusion. Importantly, our study is the first to propose that ILS is involved in chr2 fusion and may have facilitated great ape speciation and the origins of humans.

Meanwhile, while our study demonstrates that depletion of the fusion site leads to gene expression changes in NPC cells, the underlying molecular mechanisms—such as potential alterations in 3D genome structure or functional roles of genes within the fusion site—remain to be elucidated.

### RESOURCE AVAILABILITY

#### Lead contact
Further information and requests for data should be directed to and will be fulfilled by the lead contact, Yafei Mao (yafmao@sjtu.edu.cn).

#### Materials availability
This study generated wild-type and fusion-site-depletion NPCs from CN1 iPSC cell lines. Requests should be directed to Yafei Mao (yafmao@sjtu.edu.cn).

#### Data and code availability
- The T2T primate genomes used in this study are available from GenBank via the following accessions: GCA_009914755.4, GCA_028858775.2, GCA_028885625.2, GCA_028885655.2, GCA_029281585.2, GCA_029289425.2, and GCA_037993035.1. The T2T primate genome assemblies are also available on GitHub (https://github.com/marbl/Primates and https://github.com/zhang-shilong/T2T-MFA8). The Neanderthal and Denisovan genomes used are available from https://www.eva.mpg.de/genetics/genome-projects. The Iso-Seq data are available on ENCODE (tissues: ENCFF492BYP, ENCFF306ZPP, ENCFF318SKH, and CHM13-T2T hTERT Iso-Seq: SRR12519035 and SRR12519036).
- This study did not generate any unique code.

### ACKNOWLEDGMENTS

We thank Tonia Brown for editing this manuscript. We thank the HPRC and Primate T2T Consortium for providing the long-read human and great ape genome assemblies. This work was supported, in part, by grants from the National Natural Science Foundation of China (32370658); the Natural Science Foundation of Chongqing, China (CSTB2024NSCQ-JQX0004); the Shanghai Jiao Tong University 2030 Initiative (WH510363003/016); the Computational Biology Program (24JS2840300) of the Science and Technology Commission of Shanghai Municipality (STCSM); the SJTU Global Initiative Fund (Type B); Yongxin Youth Award Fund; Zhongying Young Scholar Program; and the New Cornerstone Science Foundation through the XPLORER PRIZE to Y.M. This work was supported, in part, by grants from the National Key Research and Development Program of China (2022YFF0710901), a National Natural Science Foundation of China grant (82021001), the Biological Resources Program of the Chinese Academy of Sciences (KFJ-BRP-005), and the National Science and Technology Innovation 2030 Major Program (2021ZD0200900) to Q.S. This work is partially sponsored by the Shanghai Rising-Star Program (24YF2721800) to K.M. Research reported in this publication was supported, in part, by the National Human Genome Research Institute of the US National Institutes of Health (NIH) under Award Number R01HG002385 to E.E.E. The content is solely the responsibility of the authors and does not necessarily represent the official views of the NIH. E.E.E. is an investigator of the Howard Hughes Medical Institute. This article is subject to HHMI's Open Access to Publications policy. HHMI lab heads have previously granted a nonexclusive CC BY 4.0 license to the public and a sublicensable license to HHMI in their research articles. Pursuant to those licenses, the author accepted manuscript of this article can be made freely available under a CC BY 4.0 license immediately upon publication. The computations in this study were run on the Siyuan-1, supported by the Center for High Performance Computing at Shanghai Jiao Tong University.

### AUTHOR CONTRIBUTIONS

Y.M., E.E.E., and Q.S. conceived the project; Z.Y., X.J., X.Y., K.M., S.Z., J.C., J.H., L.F., J.Z., M.V., and Y.M. contributed to the syntenic comparison, fusion site characterization, ILS, and subtelomeric repetitive region analyses; Z.Y., L.Z., X.J., Y.L., Y.N., X.B., and Q.S. contributed to the fusion site depletion analysis; D.Y. and E.E.E. generated the genome assemblies of great apes; X.J., G.Z., and Y.M. analyzed the centromeres; Y.M. and E.E.E. wrote the draft manuscript with contributions from the other authors; and all authors read and approved the manuscript.

### DECLARATION OF INTERESTS

E.E.E. is a scientific advisory board (SAB) member of Variant Bio, Inc., and a member of the *Cell* advisory board.

### STAR★METHODS

Detailed methods are provided in the online version of this paper and include the following:

## SUPPLEMENTAL INFORMATION

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

## STAR★METHODS

### KEY RESOURCES TABLE

| REAGENT or RESOURCE | SOURCE | IDENTIFIER |
|---|---|---|
| **Chemicals, peptides, and recombinant proteins** | | |
| mTeSR™ Plus | Stemcell | Cat#100-0275 |
| TrypLE | Gibco | Cat#12604-021 |
| ROCK inhibitor Y27632 | Stemcell | Cat#72304 |
| puromycin | Stemcell | Cat#73342 |
| **Critical commercial assays** | | |
| P3 Primary Cell Nucleofection Kit | Lonza | Cat#V4XP-3024 |
| TIANamp genomic DNA Kit | TIANGEN | DP304-02 |
| Sanger sequencing | Tsingke; Shanghai | https://www.tsingke.com.cn/sanger |
| STEMdiff neural progenitor medium | Stemcell | Cat#05833 |
| Hieff NGS Ultima Dual-mode mRNA Library Prep Kit | Yeasen | https://seas.ysbuy.com/prod/doc/mainDoc/12308ES-Hieff%20NGS%E2%84%A2%20Ultima%20Dual-mode%20RNA%20Library%20Prep%20Kit-Ver.EN20230327.pdf |
| **Deposited data** | | |
| Human reference genome T2T-CHM13v1.0 | T2T Consortium | GCA_009914755.4 |
| Human population genome from HPRC | Human Pangenome Reference Consortium, Liao et al.[44] | https://data.humanpangenome.org/assemblies |
| Nonhuman primates T2T genome | Yoo et al.,[38] Zhang et al.[39] | GCA_009914755.4, GCA_028858775.2, GCA_028885625.2, GCA_028885655.2, GCA_029281585.2, GCA_029289425.2, GCA_037993035.1 |
| Neanderthal and Denisovan genomes | Max Planck Institute for Evolutionary Anthropology | https://www.eva.mpg.de/genetics/genome-projects |
| **Experimental models: Cell lines** | | |
| Han Chinese CN1 cell line | Yang et al.[56] | N/A |
| **Oligonucleotides** | | |
| L-sgRNA: TGGATCGTAATCACTCGCGG | This paper | N/A |
| R-sgRNA: GGGACCTCCTCACTTGATTG | This paper | N/A |
| PCR primer F1: CCCGTGTGAGATGGTCCTTA | This paper | N/A |
| PCR primer R1: CCACTGCAGTCCGCAGTCTG | This paper | N/A |
| PCR primer F2: GCAGTATAGTGGTGGCATGC | This paper | N/A |
| PCR primer R2: TGCAGGTATGAAAATCGCCG | This paper | N/A |
| **Recombinant DNA** | | |
| BbsI-digested PX459 | Addgene | plasmid #48139 |
| **Software and algorithms** | | |
| minimap2 | Li et al.[66] | https://github.com/lh3/minimap2 |
| SafFire | N/A | https://mrvollger.github.io/SafFire |
| RepeatMasker | Dfam Consortium | https://www.repeatmasker.org/ |
| VCFtools | Danecek et al.[67] | https://vcftools.github.io/index.html |
| MAFFT | Katoh et al.[68] | https://mafft.cbrc.jp/alignment/server/index.html |
| trimAL | Capella-Gutiérrez et al.[69] | https://github.com/inab/trimal |
| iqtree | Minh et al.[70] | http://www.iqtree.org/ |
| BEAST | Bouckaert et al.[71] | https://beast.community/ |
| transanno | OKAMURA, Yasunobu | https://github.com/informationsea/transanno |
| ModDotPlot | Sweeten et al.[72] | https://github.com/marbl/ModDotPlot |

*(Continued on next page)*

CellPress

**Continued**

| REAGENT or RESOURCE | SOURCE | IDENTIFIER |
| --- | --- | --- |
| SEDEF | Numanagić et al.[73] | https://github.com/vpc-ccg/sedef |
| bedtools | Quinlan et al.[74] | https://bedtools.readthedocs.io/en/latest/ |
| circlize | Gu et al.[75] | https://jokergoo.github.io/circlize/ |
| dendextend | Galili et al.[76] | https://github.com/talgalili/dendextend |
| TreeDist | Smith et al.[77] | https://github.com/ms609/TreeDist |
| Hum-AS-HMMER for AnVIL | Fedor Ryabov | https://github.com/fedorrik/HumAS-HMMER_for_AnVIL |
| OWM-SF | Zhang et al.[39] | https://www.nature.com/articles/s41586-025-08596-w#Sec21 |
| Assembly_HSat2and3_v2.pl | Altemose et al.[78] | https://github.com/altemose/chm13_hsat |
| mrsFAST | Hach et al.[79] | https://github.com/sfu-compbio/mrsfast |
| RNA-seq pipeline | Harshil Patel et al.[80] | https://nf-co.re/rnaseq/3.19.0/ |
| DESeq2 | Love et al.[81] | https://github.com/thelovelab/DESeq2 |
| edgeR | Chen et al.[82] | https://bioconductor.org/packages/devel/bioc/html/edgeR.html |
| clusterProfiler | Yu et al.[83] | https://github.com/YuLab-SMU/clusterProfiler |

## METHOD DETAILS

### Data resources and comparative analysis

The great ape and macaque T2T genome assemblies are publicly available (Data and code availability). Syntenic relationships among primate chr2/chr2a/chr2b were assessed using minimap2[66] (v2.24) with the following parameters: '-c -x asm20 –secondary = no –eqx -Y -K 8G -s 1000' and visualized using SafFire (https://mrvollger.github.io/SafFire). We identified centromeres, transposons, and subtelomeric satellites based on RepeatMasker (v4.1.4) annotation (https://www.repeatmasker.org/).

### Fusion site characterization and population analysis

To precisely define the fusion site, we aligned NHP chr2a and chr2b to human chr2 using minimap2 (v2.24) with the following parameters '-x asm20 -r500,20000 -s 2000 -p 0.01 -N 1000 –cs'. The alignments were visualized using the minimiro (https://github.com/mrvollger/minimiro). We used VCFtools[67] (v0.1.16) to calculate nucleotide diversity (pi) and Tajima's D for population genetic analyses with the following parameters '–window-pi 20000 –window-pi-step 10000' and '–TajimaD 20000'.

### SD annotations and phylogenetic analysis

SD annotations of the fusion site in the T2T-CHM13v2.0 were based on the UCSC SEDEF-SD track (https://genome.ucsc.edu/). In this study, we focused on the SDs with an identity of ≥98% and length ≥20 kbp. These SD sequences were aligned to NHP T2T genomes to identify homologous regions using minimap2 (v2.24) with the parameters: '-cx asm20 -p 0.5 –eqx'. Orthologous segments were identified based on flanking region synteny. We aligned all primate homologous SDs with MAFFT[68] (v7.515) and used trimAl[69] (v1.4) to remove the noise sequences with the parameter: '-automated1'. The phylogenetic trees were constructed with IQ-TREE[70] (v2.1.4), and then we used BEAST[71] (v2.6.6) with the HKY model incorporating gamma site, calibrated yule, and relaxed log-normal clock models to infer the split time. Node ages were estimated using log-normal priors (Tables S6D and S6E). We conducted three independent runs for each tree and the results were consistent. Effective sample sizes exceeded 200 for all parameters in all runs.

For ILS analysis, we first truncated the fusion site flanking regions into 500 bp windows. Then, we utilized transanno (v0.4.5) (https://github.com/informationsea/transanno) to align these 500 bp segments to NHP genomes and generated multiple alignments across primate species. We then utilized IQ-TREE[70] (v1.6.12) with HKY model and ete3 python package to analyze phylogeny trees, as described previously.[37]

### Subtelomeric repetitive region characterization, methylation analysis, and TE annotation

The pairwise identity heatmaps of each subtelomeric region were generated by ModDotPlot[72] (https://github.com/marbl/ModDotPlot). We used SEDEF[73] (v1.1r35) to annotate SDs. To analyze 5mC methylation levels in pCht regions and SD spacers, we utilized BEDTools[74] (v2.30.0) to calculate mean methylation levels within 1 kbp windows, using a 500 bp step size. Synteny between subtelomeric repetitive regions was defined using minimap2 (v2.24) with the following parameters '-cx asm20 –secondary = no -A1 -B2 -O2,12 -s 1000 -Y -K 8G –eqx'. The syntenic relationship was visualized using R package circlize[75] (v0.4.16). R package dendextend (v1.18.1)[76] and TreeDist (v2.9.1)[77] were utilized to plot the co-phylogeny and to calculate the generalized Robinson–Foulds distance.

## Centromere analysis

To analyze the structure of each chromosome, we first ran RepeatMasker (v4.1.4) on all chromosomes and identified the α-satellite-enriched regions as centromeres.[78] Then, we utilized HumAS-HMMER (https://github.com/fedorrik/HumAS-HMMER_for_AnVIL) to classify the suprachromosomal families (SFs) of α-satellites in human, *Pan*, gorilla, and orangutan. For macaque, the OWM-SF annotation tool was used to annotate the SFs.[39] The HOR arrays were characterized using the tool StV (https://github.com/fedorrik/stv). However, for orangutan chr2a and chr2b, the tool encountered difficulties in correctly identifying the HORs, likely due to the complex structure of these acrocentric chromosomes. Thus, we selected the largest continuous regions containing the same SFs as their HORs. We then estimated the frequency of 5mC and CpG methylation within the HOR regions. BEDTools (v2.30.0) was used to count the frequency of methylation within 5 kbp windows and we determined the regions with the minimum frequency and below the lower quartile among the whole HOR as CDRs. Additionally, we used the previous published tool[84] (https://github.com/altemose/chm13_hsat) to identity the HSatII/HSatIII arrays in human and *Pan* peri/centromeric regions (α-satellite-enriched regions and 5 Mbp on the p-arm and q-arm).

We further characterized the human centromere degenerate site through synteny comparisons between *Pan* and human, incorporating RepeatMasker annotations of 'ALR_Alpha.' Breakpoints between chimpanzees and humans were identified through alignment and analysis of specific SF organizations. ModDotPlot (https://github.com/marbl/ModDotPlot) was used to generate heatmaps for chimpanzee chr2b with a window size of 5,000 bp and for the human centromere degenerate site with a window size of 200 bp.

To assess the diversity of centromere degeneration across human populations, we first confirmed the presence of this site in all human genomes using minimap2 (v2.24) with the parameters '-cx asm20 –secondary = no -s 2500'. We then extracted the targeted regions from each assembly from HPRC data and ran RepeatMasker (v4.1.4) on these regions. We defined three satellite arrays—cenD_1, cenD_2, and cenD_3—with subtypes determined by length variations.

To confirm the fusion occurred in archaic humans, we randomly selected two individuals from the five structural haplotypes (total 10 individuals) and five NHP genomes to run mrsFAST[79] (v3.4.2) for identifying singly unique nucleotide *k*-mers (SUNKs) in modern humans. Subsequently, we checked the counts of the SUNKs in archaic human genomes.

To compare the methylation frequencies among human chr2, α-satellite arrays at the degenerate centromeric site, and NHP centromeres, we chunk the HORs (excluding CDRs; for macaques, we used the whole α-satellite region, excluding CDRs) into 17.1 kbp windows by BEDTools (v2.30.0) with parameters: '-w 171000 -s 8550' and calculate the frequencies within windows. Visualizations were generated using ggplot2.

## Fusion site KO experiments

The design of optimal sgRNA pairs to target sites was performed using the online CRISPR design tool (http://crispor.gi.ucsc.edu/). The complementary oligonucleotide pairs of L-sgRNA and R-sgRNA were annealed at 95°C for 5 min, with ramp-down to 25°C to generate the double-stranded DNA (dsDNA) fragment, before ligation into BbsI-digested PX459 (plasmid #48139, Addgene).

Wild-type induced pluripotent stem cells (CN1) or fusion site depletion induced pluripotent stem cells were maintained in mTeSR Plus (Stemcell). Cells were dissociated into single-cell suspensions using TrypLE (Gibco) and electroporated using the Lonza Nucleofector 4D X Unit (program CA137) and the P3 Primary Cell Nucleofection Kit (V4XP-3024). The following conditions were used: 5–6 × 10^5 cells/mL, 10 μg sgRNA (two sgRNAs mixed at a 1:1 ratio). The control groups were electroporated with scramble sgRNAs. After electroporation, iPSCs were resuspended in mTeSR Plus supplemented with 10 μM ROCK inhibitor Y27632 and plated onto 12-well culture plates within 24 h. At 48 h post-electroporation, the culture medium was replaced with fresh mTeSR Plus containing puromycin (1μg/mL) for 48h of selection. Cells were subsequently maintained in standard mTeSR Plus medium until colony formation.

We selected three independent monoclonal iPSC-CN1 cell lines for the knockout (KO, *n* = 3) and control (CTRL, *n* = 3) condition. For isolation of KO or control monoclonal clones, selected colonies were dissociated into single-cell suspensions. Individual cell was manually transferred using a mouth pipette under a stereomicroscope and seeded into 96-well plates. When a single cell grows into a clonal colony, we extract the genomic DNA from the cells with the TIANamp genomic DNA Kit (DP304-02, Tiangen) according to the manufacturer's instructions. Genome deletion was detected by PCR-amplification of gDNA using a primer pair flanking the deletion. The primers used for PCR screening are listed in Table S5A. Genomic PCR products were detected by agarose gel electrophoresis and Sanger sequencing (Shanghai, Tsingke).

## iPSCs differentiation into neural progenitor cells

Neural Progenitor Cells (NPCs) were generated from control iPSCs or KO iPSCs using the STEMdiff Neural System (Stemcell). Briefly, Cells were harvested using TrypLE and resuspension in STEMdiff Neural Induction Medium + SMADi +10 μM Y-27632 as single cells. Add cell suspension (2 × 10^6 cells/well) to a single well of the matrix-coated 6-well plate and then perform a daily full-medium change with warm (37°C) medium until cultures are ready to be passaged. At passage 3, NPCs were expanded in STEMdiff Neural Progenitor Medium.

### RNA preparation and RNA-seq data analysis

Total RNA was isolated from each replicate of the KO cell lines and the control cell lines using Hieff NGS Ultima Dual-mode mRNA Library Prep Kit, following the manufacturer's instructions. We utilized RNA-seq pipeline[80] from the nf-core[85] community with default parameters to align and quantify the RNA-seq data. Differential expression analysis was conducted using the R package DESeq2[81] and edgeR.[82] We identified differentially expressed genes (DEGs) with following criteria: false discovery rate (FDR) < 0.05 and a $\log_2$ fold change >2. Because edgeR and DESeq2 produced similar results, we used the results by edgeR to do the further analysis. We performed Gene Ontology (GO) enrichment analysis using clusterProfiler.[83]

