## [Document S2. Transparent peer review records for Yang et al. · Cell Genomics]

Incomplete lineage sorting of segmental duplications defines the human chromosome 2 fusion site early during African great ape speciation

Zikun Yang, Lu Zhang, Xinrui Jiang, Xiangyu Yang, Kaiyue Ma, DongAhn Yoo, Yong Lu, Shilong Zhang, Jieyi Chen, Yanhong Nie, Xinyan Bian, Junmin Han, Lianting Fu, Juan Zhang, Mario Ventura, Guojie Zhang, Qiang Sun, Evan E. Eichler, Yafei Mao

Summary

Initial submission: Received : Jan 14, 2025

Scientific editor: Sara Rohban

First round of review: Number of reviewers: 3
Revision invited : Feb 24, 2025
Revision received : Jul 15, 2025

Second round of review: Number of reviewers: 3
Revision invited : Aug 5, 2025
Revision received : Aug 10, 2025

Third round of review: Number of reviewers: 1
Accepted : Nov 3, 2025

Data freely available: YES

Code freely available: YES

This transparent peer review record is not systematically proofread, type-set, or edited. Special characters, formatting, and equations may fail to render properly. Standard procedural text within the editor's letters has been deleted for the sake of brevity, but all official correspondence specific to the manuscript has been preserved.

Referees' reports, first round of review

Reviewer #1:

This study leverages recently assembled telomere-to-telomere (T2T) genomes for humans and non-human primates (NHPs) to characterize (in greater detail than previously possible) the formation of human chr2 via the fusion of NHP chr2a and chr2b. The key findings are a base pair level description of the fusion site, evidence that chr2a and chr2b telomeres evolved via incomplete lineage sorting, and dating of the fusion event. Editing out the fusion site in human HEK293T cells revealed some differentially expressed genes enriched for transcriptional regulators, though the mechanism and consequence of differential expression are not explored.

- 1) It would be helpful to better highlight what data in this study and what results depend on the T2T assemblies. By clarifying what could have been done previously (or was previously known) versus what is only possible now, and what conclusions would have been wrong if the analyses had used pre-T2T genomes, the timeliness and novelty of the findings will be more evident.
- 2) Only readers with deep expertise in segmental duplications (SDs), satellites and other repetitive DNAs, and chromosome evolution will easily follow the narrative in the results section. To help other readers understand the significance of the analyses, it would be useful to include some prose about each type of sequence variant analyzed and how they typically evolve. For instance, it is pretty easy to see how homologous repeats at the ends of two chromosomes could facilitate fusion (Figure 6b first step on human branch), but it may be less obvious why SD_fusions A-C, present on a variety of chromosomes, would localize at the fusion site either before (Figure 6a) or after (Figure 6b) the fusion event. Separately talking about the subsequences that directly mediate fusion versus the rest of the dynamic evolution of the subtelomeres / fusion site might be useful. Overall, it will be important to also discuss what evolutionary events at the fusion site are typical for the various types of sequence elements and which, if any, are particularly surprising.
- 3) Similarly, the results section would benefit from an opening subsection describing the different types of analyses that will be performed and what each one is able to demonstrate or test. This will help readers recognize and understand what is being investigated in each of the following subsections (headers provide sign posts, but are too vague to enable a general reader to queue up what kind of analytical methods and data to expect to see in the section). Additional introductory and concluding sentences in each results subsection are also recommended to underscore what question is asked and what answer is revealed. For example, you could use a formulation like "If X happened, we would see Y in the data, whereas if we saw B in the data, we would conclude that A happened.... These findings show B, and therefore A is the most likely mechanism for....".
- 4) The caption of Figure 3 uses the word "rapid", but the basis for this description is not clear. Was there a test performed to assess a faster than neutral rate?
- 5) Testing edited cell lines for differentially expressed genes (DEGs) is challenging, so it is appreciated that three replicates for each of two edits were assessed. Nonetheless, the enrichment for transcription factors (TFs) amongst the DEGs is a little bit concerning, since many TFs are low expressed and hence may show elevated expression variability between replicates. Very highly expressed genes can also be variable. Variability can lead to spurious DEG results. To address this, the authors could examine the average expression level of the DEGs versus other genes. If their levels are typical, then this is less of a concern. Another important validation is investigating the role of multi-mapping reads in the RNA-seq

analyses. How were multi-mapping reads handled and is it possible that in the WT control the deleted region transcripts are stealing reads from other genomic locations that are then mapped to these locations when the deleted transcript are removed from the alignment target? Regardless, it would be helpful to think about 5/10 DEGs validating by RT-PCR from a statistical perspective: is this an expected validation rate for false positives or indicative of many DEGs being true positives?

6) The DEG results would be more impactful if the mechanism of differential expression and the potential consequences were explored a bit more deeply. Is transcription (or lack of transcription) of the (pseudo)genes in the SDs involved or not? Were any of the DEGs in other copies of SD_fusion_A, B or C on other chromosomes? Does removing the fusion region lead to generalized stress response?

7) Given the estimated timing of the fusion event and high degree of ILS, what role do you think that the emergence of human chr2 played in speciation?

Reviewer #2:

In this manuscript, the authors report an extensive analysis of the structure and evolution of the fusion site on human chromosome 2. It has been known for many years that the largest karyotypic difference between humans and our great ape relatives is the unique fusion of two ancestral chromosomes to form modern human chromosome 2. The fusion site and a decayed centromere were mapped and described years ago. But due to the complex and highly repetitive structure of the fusion site on human chromosome 2, and the equally complex and repetitive nature of the relevant regions in the chimpanzee and gorilla chromosomes, it has been impossible to resolve in any detail the structure of the fusion site in humans, or the homologous structures in the apes.

Jiang et al. report here the use of the recently produced T2T genome assemblies for human, chimpanzee, bonobo, gorilla, two orangutan species and rhesus macaque to investigate the content and evolution of the fusion site. The manuscript describes in impressive detail and laudable clarity the complex structures of the relevant regions in the African apes and humans. The text provides a thorough explanation of the relevant telomeric and subtelomeric regions from the various species, as well as defining and characterizing the multiple copies of segmental duplications both within each single species genome and across species. The figures present useful diagrams of the relationships among the various chromosomal regions and outline the inferred evolutionary steps that led to both the differentiation of chromosome 2a and 2b content across the great apes and the evolutionary origin of the structure of the fusion region in humans.

I find the descriptions of the regions to be thorough and sufficient. The documentation of the various structural elements is sufficient to provide the reader with an appreciation for the complexity and the dynamics of the relevant chromosomal segments. The functional analysis (i.e. comparing how gene transcription in HEK293T cells differs depending on presence or depletion of the fusion site) is interesting and significant as it illustrates the potential wider effect of the chromosomal fusion event. Overall this is a valuable contribution to understanding of both the evolution of the human genome and the mechanisms and genomic details of karyotypic change more generally.

The comparisons across species indicate that there is no simple evolutionary scenario that can explain the pattern of shared and differentiated chromosomal structures across the radiation of humans, chimpanzees, gorillas and orangutans. Rather, the authors argue that incomplete lineage sorting provides the only sufficient explanation for why the structure of the human fusion region is not a simple derivative of either the chimpanzee/bonobo structures or the gorilla structures. The authors invoke incomplete lineage sorting (ILS) to explain why the present-day distributions across species of SD_fusion_A, SD_fusion_B and SD_fusion_C cannot be represented as a simple evolutionary tree but

rather imply different histories for different chromosomal segments.

In my opinion, it is possible that ILS is the explanation for this complexity, but I do not believe it is the only possible explanation. While ILS might be a sufficient mechanism, there are now many published examples of introgression and hybridization between recently diverged primate and other mammalian species. Introgression from one species into a closely related species is (a) more common than was assumed years ago, (b) a sufficient explanation for the distinct histories of the various relevant chromosomal segments and (c) a process distinct from ILS. Jiang et al. found that there are multiple degenerate haplotypes present in the human population for the centromere that became non-functional after the chromosomal fusion. It is quite possible that at various times during the initial differentiation of the human, Pan and gorilla lineages that each incipient lineage/species had several haplotypes segregating for chromosomes 2a, 2b and the fusion. Gene flow from one incipient lineage into another seems to me to be just as plausible an explanation for the complex contradictory histories of the various chromosomal sub-regions. Furthermore, there is compelling evidence for additional African great ape lineages that went extinct, the so called ghost lineages (e.g. Pawar et al PMID37500909 and Kuhlwilm et al. PMID31036897). The authors should either explain why it is not possible that during the early evolutionary differentiation of the extant lineages (periods when there were increasing genetic differences but still opportunities for inter-lineage gene flow) there could not have been introgression of distinct haplotypes from one lineage into another, involving either two of the lineages that survived to the present or one of those and a now extinct ghost lineage. It seems to me that the difference in ILS proportions proximal and distal to the fusion site provides some evidence that it was not just ILS that generated the complex pattern the authors reveal.

In Line 323 the authors speculate that the fusion event had the effect of "...creating a stasipatric speciation barrier." This is also debatable. If the authors are implying an immediate barrier to inter-breeding arose between the population in which the fusion occurred and the other closely related lineages, this is not reasonable. The fusion event would (most likely) have occurred only one time and heterozygotes for the ancestral condition and the fusion must have been fertile for many generations while the fused chromosome increased in frequency. This presumed fertility of heterozygotes means an immediate barrier to inter-population breeding is very improbable. The authors will have to provide much stronger argument if they wish to infer a direct connection between the fusion event and "speciation."

Lesser concerns: There are a number of minor issues in the manuscript that should be addressed.

Lines 117-118: I believe this should read "...while another occurred after the Pan-gorilla split."

Line 148 and after: The authors suggest that the date of the divergence of the gorilla lineage from the human-Pan lineage was about 7.4 mya. My reading of the literature suggests that the date should be older than that. Both Shao et al. (Science 380: 913, 2023) and Dos Reis et al. (Syst. Biol. 67: 594, 2018) put it at least 9.9 mya. Other papers also suggest a date older than 7.4 mya. The authors may have strong justification for the younger date. But if they do, they should cite the supporting prior studies.

Lines 321-322: I think the authors intend to say: "...we speculate that this fusion event did NOT arise in the genus Homo...."

Minor issues:

Line 58: Reference #7 is listed in the reference list as Painter and Stone, not Wilson and Painter.

Line 191: Given how the field today is emphasizing telomere-to-telomere (T2T) assemblies, it might be better to drop that use of the T2T abbreviation for a different concept. I do not think that abbreviation is used again in the manuscript.

Reviewer #3:

In this paper the authors use the completed great ape sequences (to be published in another paper) to reconstruct the genomic events that led to the fusion of human chromosome 2. This is a complicated (and wonderfully nerdy :-)) endeavor given the complexity and details of the reconstruction of the duplicated, inverted, rearranged and twisted SD regions. It is also as the authors rightly say " arguably the most significant karyotypic difference between humans and NHPs" and hence in my opinion absolutely justified, interesting and relevant to get the region and its evolutionary history straight. Together with the follow-up on the transcriptional effects of the deletion, I think that this is a very good and adequate paper for Cell Genomics.

As a disclaimer, I cannot fully judge the correctness of all details of the complex reconstructions, but given the expertise of the authors and the presented results, this makes a very reasonable impression. This said, I have two major issues that should be improved before publication

A) Clarity of text

While I find the Introduction very clear and understandable, I think parts of the paper including the discussion can profit from a more clear and understandable writing. It is beyond the scope of my review to suggest rewritings, but maybe if the person who wrote the introduction could improve also the other parts, the manuscript and hence authors, readers, journal would profit quite some bit. Here are a few minor points I came across:

L47-48: "Most conspicuously, one of these SDs shares homology to a hypomethylated SD spacer sequence present in hundreds of copies in the subterminal heterochromatin of chimpanzees and bonobos" Its not clear to me in what way this is conspicuous, i.e. which mechanism/process/evolution/hypothesis is informed by this finding.

L51-52: change to "...alters the expression of genes in HEK cells" as I think it is important that this is in HEK cells

L55: what is meant by "with ILS in speciation"? A bit unclear sentence, especially for the last one of the abstract..

L74 difference (singular) not differences?

L154: I think when ILS first appears, one should quickly remind reads what that is and maybe give a reference where they can read it up, as I am not sure that all of the readers interested in the article will be familiar with this phenomenon

L 322: should be:" ... we speculate that this fusion event did not arise in the genus Homo..."

B) Fusion site depletion

I think the depletion of the fusion site and the following RNA-seq is really a highlight of the paper as it goes beyond the pure sequence reconstruction and the most important biological result from the paper. But I think the experiment is not described sufficiently and also not replicated sufficiently:

1) A better description of what was actually done is needed. The method section is unclear how exactly cells were selected and it is very unclear how RNA-seq was done (it is just mentioned that total RNA was reverse transcribed and normalized to GAPDH, but this refers to the QPCR, I guess!). To be honest, the description does not suggest a lot of experience with RNA-seq analyses as for example the amount of produced Gbp is mentioned in the main text, but what matters and is usually reported is the number of reads. It is also very uncommon to use two statistical programs (limma and edgeR) and take select genes that are significant in both. Furthermore, QPCR from the same clones does not hurt, but is not thought to be actually necessary as QPCR is less sensitive than properly performed RNA-seq. However, what is of great importance is the type of replicates and it is not clear from the text how lines of wt and modified HEK cells were chosen and hence it is unclear what was replicated.

2) The central claim is that the deletion causes expression changes in HEK cells. To be valid, this needs to be significant, i.e. needs to be found in independent replications of this experiment. As the description of what was done is not clear (see above), I assume that 3 wells from one single-cell isolate of sg2, one from sg1 and culture from the wildtype ancestors was done. What is relevant is that the heterogeneity in HEK cells (or any cell culture) in combination with single cell cloning can lead to substantial differences in expression that are not related to the genetic manipulation performed but due to genetic and epigenetic properties of the single clone. Notably, the relatively small overlap between the two lines sg1 and sg2 suggests this to be the case (although difficult to judge given the limited description of the experiments). I think to really nail the point that the fusion causes expression changes, one would need to make RNA-seq of at least six single-cell cloned HEK lines that carry the deletion (either sg1 or sg2, assuming that they have identical effects) and six that did not carry it (i.e. WT clones that underwent the same procedure e.g. with scrambled sgRNAs or at a minimum derived from a single clone of the same population). Note that I find the proper design of the experiment the crucial point and not the result. The analysis is as interesting to be published also if one finds no DEGs

3) HEK cells are actually not kidney epithelial cells, but probably adrenal cells of neuronal origin (not sure, but I think this is the proper reference: <https://pmc.ncbi.nlm.nih.gov/articles/PMC4166678/>). Not that this would devalue the experiments per se, but I think it is important mentioning.

4) HEK cells are also known to have massive genomic rearrangements; Given the topic of the paper, I think it would be important to address how many copies of the fusion site are actually present in HEK cells

Authors' response to the first round of review

Reviewer comments are shown in **black** text, with our responses in **blue**. All changes to the manuscript are highlighted in **yellow** in the file titled “**main_text_revised_clean**,” with line numbers referenced in the rebuttal corresponding to this version. For reference, we have also provided a tracked-changes version of the manuscript (“**main_text_revised_tracked-changes**”).

Reviewer #1:

This study leverages recently assembled telomere-to-telomere (T2T) genomes for humans and non-human primates (NHPs) to characterize (in greater detail than previously possible) the formation of human chr2 via the fusion of NHP chr2a and chr2b. The key findings are a base pair level description of the fusion site, evidence that chr2a and chr2b telomeres evolved via incomplete lineage sorting, and dating of the fusion event. Editing out the fusion site in human HEK293T cells revealed some differentially expressed genes enriched for transcriptional regulators, though the mechanism and consequence of differential expression are not explored.

Thank you for your time in carefully reviewing our manuscript.

1) It would be helpful to better highlight what data in this study and what results depend on the T2T assemblies. By clarifying what could have been done previously (or was previously known) versus what is only possible now, and what conclusions would have been wrong if the analyses had used pre-T2T genomes, the timeliness and novelty of the findings will be more evident.

This is a good suggestion. First, we made a table that includes the genome information used in this study with detailed genomic quality assessment values (Supplementary Table 13 and 14).

Supplementary Table 13. Summary of T2T genomes.

Species	NCBI RefSeq assembly	Contig N50 (Mbp)	QV
Homo sapiens	GCF_009914755.1	150.6	75.9
Pan troglodytes	GCF_028858775.2	146.3	66.0
Pan paniscus	GCF_029289425.2	147.0	62.7
Gorilla gorilla	GCF_029281585.2	151.4	61.7
Pongo pygmaeus	GCF_028885625.2	140.6	65.8
Pongo abelii	GCF_028885655.2	146.2	63.3
Macaca fascicularis	GCF_037993035.1	162.1	71.4

Supplementary Table 14. Summary of T2T chr2/chr2a/chr2b.

Species		chr2a / bp	chr2b / bp	chr2a QV	chr2b QV
Homo sapiens		242,696,752		80.1	
Pan troglodytes	hap1	122,853,238	144,476,839	75.6	80.5
	hap2	126,935,857	140,842,815	65.2	73.2
Pan paniscus	mat	126,551,469	144,872,268	71.6	78.1
	pat	123,784,342	147,478,900	70.4	70.1
Gorilla gorilla	mat	151,425,642	150,248,535	77.5	63.7
	pat	145,906,006	144,025,524	69.6	71.8
Pongo pygmaeus	hap1	134,313,182	143,261,986	62.7	77.5
	hap2	137,696,483	141,980,120	68.6	73.0
Pongo abelii	hap1	133,183,124	146,197,178	74.6	79.4
	hap2	131,851,162	145,714,809	66.2	71.8
Macaca fascicularis		122,129,872	138,599,521	69.1	81.4

Second, we focused on fusion sites, centromeres, and subtelomeric repetitive sequences in NHP genomes. By comparing the recent NHP T2T genome with previous long-read NHP genomes (different species), we observed that centromeres and subtelomeric heterochromatic regions (e.g., pCht) were not resolved in any of the earlier NHP genome assemblies (Supplementary Figure 27).

Supplementary Figure 27. The chr2a/chr2b syntenic comparisons between previous genomes and T2T genomes in NHPs. The blue blocks show the syntenic alignments while the yellow blocks represent the inverted alignments. The red and pink blocks represent the previous unresolved regions.

In addition, we provided more detailed analysis of the subtelomeric regions, including the SD_fusion_A/B/C on NHP genomes. We found 63 out of 74 SDs homologous to the human fusion site in T2T NHP genomes were not resolved in previous NHP genome assemblies (Supplementary Table 15).

Supplementary Table 15. The comparison of human fusion site homologous regions between T2T NHP genomes and previous NHP long-read genomes.

Species	T2T_Chromosome	T2T_Start	T2T_End	Pre** Chr	Pre** Start	Pre** End	Comment
SD_fusion_A (chr2: 113,940,058-113,990,477)							
chimpanzee	chr11_hap1_hsa9	109791	149996				
	chr11_hap1_hsa9	58849430	58887932				
	chr13_hap1_hsa2b	144292328	144332538				
	chr17_hap1_hsa18	90575949	90614548				
	chr2_hap1_hsa3	201408063	201457937				
	chr21_hap1_hsa20	6703203	6745591				
	chr4_hap1_hsa5	675584	714178				unresolved
	chr4_hap1_hsa5	178818566	178858763				
	chr5_hap1_hsa6	6067693	6107701				
	chr5_hap1_hsa6	182799661	182838475				
	chr7_hap1_hsa8	155522294	155572127				
	chrX_hap1_hsaX	153413825	153452404				
bonobo	chr1_pat_hsa1	227528048	227566495				
	chr10_mat_hsa12	141648230	141689636				
	chr11_mat_hsa9	110530	14929				
	chr11_mat_hsa9	1040051	1078462				
	chr11_mat_hsa9	67535680	67574193				
	chr11_mat_hsa9	124600602	124639029				
	chr13_mat_hsa2b	144681470	144721843				
	chr14_pat_hsa13	115943099	115981469				
	chr15_mat_hsa14	108321616	108360202				
	chr18_pat_hsa16	93838	132278				
	chr18_pat_hsa16	100367356	100405749				
	chr2_mat_hsa3	196163740	196202330				
	chr21_mat_hsa20	842296	884063				unresolved
	chr21_mat_hsa20	1059215	1101822				
	chr3_pat_hsa4	188448426	188488828				
	chr4_mat_hsa5	111923	150349				
	chr4_mat_hsa5	176386997	176425072				
	chr5_pat_hsa6	14691845	14729968				
	chr5_pat_hsa6	190989685	191028079				
	chr6_mat_hsa7	195945566	195992983				
chr7_pat_hsa8	162635258	162673656					
chr8_mat_hsa10	145926308	145964770					
chr8_mat_hsa10	146851936	146890335					
chrX_mat_hsaX	156001814	156041837					
gorilla	chr11_mat_hsa2b	15063923	15102463				unresolved
	chr12_pat_hsa2a	140637826	140676353				
	chr13_pat_hsa9	64547109	64596841				contig in chr9:39449163-39465542
	chr3_pat_hsa4	212329667	212368215				
	chr4_pat_hsa17x5	185286156	185324094				unresolved
	chr7_pat_hsa8	151215077	151253464				
S. orangutan	chrX_mat_hsaX	167752401	167790875				
	chr13_hap1_hsa9	58705971	58744603	chr9	41198782	41237493	
macaque	chr13_hap1_hsa9	58745108	58754913				unresolved
	chr15	99033879	99038026				
	chr15	99039782	99076729	chr15	83763252	83800182	
	chr15	99077215	99088261	chr15	83800662	83811693	

For example, a homologous region of SD_fusion_A is located on chromosome 9 in NHGRI_mPanTro3, but it has not been resolved in the long-read genome of Clint_PTR (Supplementary Figure 28).

Supplementary Figure 28. The comparison of a region containing SD_fusion_A in chromosome 9 between chimpanzee T2T genome and previous genome. The red block represents the SD_fusion_A homologous region in the chimpanzee T2T genome, while it wasn't assembled in previous genome. The gray block represents the gap in this region (scaffolded by N).

Finally, we did the same comparison on different human reference genomes, including GRCh37, GRCh38, and T2T-CHM13. The fusion site and degraded centrosomes are well assembled in different reference genomes (Rebuttal Figure 1 and 2).

Rebuttal Figure 1. The gene annotation track at the fusion site in T2T-CHM13v2.0:113,940,058-114,049,496 (a), GRCh38: 113,515,526-113,624,768 (b) and GRCh37: 114,273,103-114,382,345 (c) from UCSC Genome Browser.

Rebuttal Figure 2. The repeat annotation track at the centromere degenerate site in T2TCHM13v2.0: 132,644,386-132,685,996 (a), GRCh38: 132,208,800-132,250,410 (b) and GRCh37: 132,996,373-133,007,983 (c) from UCSC Genome Browser.

Overall, while the human genomes have been well-assembled, previous NHP genomes were not as thoroughly assembled. The recent T2T NHP genome assemblies, however, provide a valuable resource for investigating the fusion event. We have added these results to our revised manuscript and added the sentence in the Discussion line 335-340:

“In previous NHP assemblies, SDs homologous to SD_fusion_A/B/C were largely collapsed (63 out of 74 cases), and subtelomeric repetitive sequences remained unresolved (Supplementary Figure 27-28 and Supplementary Table 12-14). Complete T2T sequence for chromosome 2, 2a, and 2b among the great apes (average QV=72.1) allowed us to systematically examine the complex genomic architecture and further refine the evolutionary history of the human-specific chr2 fusion event.”

2) Only readers with deep expertise in segmental duplications (SDs), satellites and other repetitive DNAs, and chromosome evolution will easily follow the narrative in the results section. To help other readers understand the significance of the analyses, it would be useful to include some prose about each type of sequence variant analyzed and how they typically evolve.

Thank you for your good suggestion. We revised our introduction and added a brief introduction on repetitive sequences study at line 101-107:

“In the absence of complete genome assemblies, most SDs and satellite arrays remain unresolved (Aganezov et al., 2022; Vollger et al., 2022), limiting our ability to examine their evolutionary dynamics. Previous studies using long-read sequencing have shown that SDs can undergo rapid lineage-specific expansion (Mao et al., 2021; Yoo et al., 2025; Zhang et al., 2025)(Mao et al., 2021; Yoo et al., 2025; Zhang et al., 2025), yet, no comprehensive analysis has been performed at the human chr2 fusion site. Meanwhile, pCht satellite sequences have been fully characterized in recent ape genome study (Yoo et al., 2025), but their detailed relationship with the fusion event remains unclear.”

For instance, it is pretty easy to see how homologous repeats at the ends of two chromosomes could facilitate fusion (Figure 6b first step on human branch), but it may be less obvious why SD_fusions A-C, present on a variety of chromosomes, would localize at the fusion site either before (Figure 6a) or after (Figure 6b) the fusion event. Separately talking about the subsequences that directly mediate fusion versus the rest of the dynamic evolution of the subtelomeres / fusion site might be useful.

Our previous text may have unintentionally led to a misunderstanding of Figure 6a and 6b. The key distinction between the two models is not about whether pCht sequences appeared before or after the fusion event. Rather, the central difference lies in which type of sequence primarily mediated the fusion—SDs or pCht repetitive sequences.

In Figure 6a, we propose that SDs played the major role in mediating the fusion. Under this model, structurally diverse subtelomeric haplotypes co-existed within the ancestral African great ape population, and one such configuration facilitated the fusion. This one could be with pCht or not. In Figure 6b, we suggest that pCht repetitive sequences were the primary mediators of the fusion event, which occurred in the ancestral human lineage. This model implies that the fusion led to the loss or reorganization of pCht sequences in the derived human lineage.

We revised our text in Discussion line 413-416:

“These two models differ in terms of which repetitive sequences played the primary role in mediating the fusion event: the first model favors SDs, while the second emphasizes pCht-like sequences. However, both models support the presence of SDs in the LCA, and suggest that ILS potentially facilitated by a large effective population size played a key role.”

In addition, we attempted to “*Separately talking about the subsequences that directly mediate fusion versus the rest of the dynamic evolution of the subtelomeres / fusion site might be useful.*”, as suggested. However, due to the complexity of these repetitive sequences’ evolutionary processes and the structure of the existing narrative, we chose to maintain the original writing logical flow. That said, we revised the text in the Discussion section in accordance with your other suggestions, and we believe the current version significantly improves clarity and readability for the audience. We appreciate your insightful comments and understanding.

Overall, it will be important to also discuss what evolutionary events at the fusion site are typical for the various types of sequence elements and which, if any, are particularly surprising.

Fusion events mediated by SDs have been reported previously (Ventura et al., 2012; Poszewiecka et al., 2023); however, fusions involving pCht sequences have not been described. Nevertheless, fusion mediated by non-allelic homologous recombination between highly identical repetitive sequences is a well-recognized genomic mechanism (Parks et al., 2015; Jeong et al., 2025). A surprising finding in our study is the evidence of ILS involving SDs at the fusion site—an observation not previously reported, even in other species. We propose that the large ancestral population size of African great apes contributed to elevated structural diversity in subtelomeric regions, potentially facilitating this fusion event. To our knowledge, this is the first study to report ILS involving complex repetitive elements such

as SDs, and to explore their role in speciation. Furthermore, while previous models suggest that the fusion occurred after the human–chimpanzee divergence (Dreszer et al., 2007; Poszewiecka et al., 2022), our findings support an alternative scenario in which the fusion may have occurred prior to the split, though its fixation time remains unclear.

We emphasized this in our discussion as you suggested at line 376-379:

“It is well established that fusion events can be mediated by SDs and other repetitive sequences (Ventura et al., 2012; Poszewiecka et al., 2023). However, to our knowledge, this study presents the first evidence of SDs subject to ILS being involved in a speciation event. Based on our observations, we propose two evolutionary scenarios for the formation of human chr2.”

3) Similarly, the results section would benefit from an opening subsection describing the different types of analyses that will be performed and what each one is able to demonstrate or test. This will help readers recognize and understand what is being investigated in each of the following subsections (headers provide sign posts, but are too vague to enable a general reader to queue up what kind of analytical methods and data to expect to see in the section). Additional introductory and concluding sentences in each results subsection are also recommended to underscore what question is asked and what answer is revealed. For example, you could use a formulation like "If X happened, we would see Y in the data, whereas if we saw B in the data, we would conclude that A happened.... These findings show B, and therefore A is the most likely mechanism for....".

Thanks for your suggestion and we revised the text as the following:

To help readers to better understand the comparative genomics analysis in primate chr2, we added an open sentence below:

Line 115-119: “To characterize the syntenic regions and structural changes associated with the fusion site in primates, we performed a comparative analysis of 10 finished nonhuman great ape chromosomes and the finished macaque genome to human chr2 (Figure 1a, Methods).”

Line 136-140: “To precisely identify the fusion site at single-base-pair resolution and to decompose its substructure, we further compared human chr2 with *Pan* chr2a and chr2b to characterize the ~109 kbp fusion site in the human T2T-CHM13v2.0 genome assembly (2q14.1, chr2:113,940,058-114,049,496) (Figure 1b and Supplementary Figure 2)”

To help readers to better understand the ILS study, we added an open sentence below:

Line 200-203: “Next, we investigated whether ILS also occurs in the regions flanking the fusion site. To this end, we extended the ILS analysis to the 5 Mbp mapping proximally (chr2a) and distally (chr2b) to the fusion site in humans (Methods) using a 500 bp windowed approach (Supplementary Table 4).”

To help readers to better understand the argument about telomeric sequences in the fusion site and if these sequences facilitated fusion, we added a guiding sentence below:

Line 210-214: “Previous investigations identified inverted telomeric sequences (TTAGGG/CCCTAA) at the fusion site, suggesting a telomere-to-telomere fusion. However, whether these sequences are remnants of the ancestral telomeres from chr2 fusion, or simply telomeric-like repeats interspersed within ancestral non-telomeric regions, remains unresolved.”

To help readers to better understand why we performed the functional analysis, we added a guiding sentence below:

Line 297-301: “As the fusion site is nearly fixed in the human genome, we speculated that it may confer a functional advantage. To explore this possibility, we examined gene models and expression patterns in this region and generated knockout (KO) cell lines to assess potential gene expression changes associated with the fusion site.”

4) The caption of Figure 3 uses the word "rapid", but the basis for this description is not clear.

Was there a test performed to assess a faster than neutral rate?

Good point. Estimating the evolutionary rate of repeats in primates is challenging, so it is inappropriate to use 'rapid' in this context. We removed the term 'rapid' to better describe the motif turnover.

5) Testing edited cell lines for differentially expressed genes (DEGs) is challenging, so it is appreciated that three replicates for each of two edits were assessed. Nonetheless, the enrichment for transcription factors (TFs) amongst the DEGs is a little bit concerning, since many TFs are low expressed and hence may show elevated expression variability between replicates. Very highly expressed genes can also be variable. Variability can lead to spurious DEG results. To address this, the authors could examine the average expression level of the DEGs versus other genes. If their levels are typical, then this is less of a concern.

Thank you for your valuable suggestions. In response to concerns raised by you and the other two reviewers regarding the KO analysis, we have spent the past four months redesigning the experiments and conducting more rigorous data analyses in accordance with your suggestions.

In this revision, we performed CRISPR-Cas9 editing using a single guide RNA pair (L-sg1&Rsg). For the control group, we added scrambled sgRNAs. After validating successful depletion, we selected three independent monoclonal iPSC-CN1 cell lines for both knockout (KO, $n = 3$) and control (CTRL, $n = 3$) cell lines and differentiated them into NPCs and performed RNA-seq on each line. RNA expression clustering revealed that KO samples grouped together, and control samples clustered separately, indicating that the knockout is the primary factor contributing to the differences between the control and KO groups (Supplementary figure 26). This revised experimental design using iPSC-CN1 offers improved genetic clarity and robustness over our initial experiments in HEK293 cells.

Supplementary Figure 26. RNA-seq analysis of CN1-derived neural progenitor cells. (b) Principal component analysis (PCA) of RNA-seq replicates using the top 500 most expressed genes shows strong replication within each condition. (c) The Euclidean distance matrix illustrates the consistency within groups and large differences between groups.

We performed immunofluorescent staining for stem cell and NPC marker genes in the cell lines used in this study. As shown below, all cell lines exhibited clear expression of the corresponding marker genes (Rebuttal figure 3 and 4).

Rebuttal Figure 3. The immunofluorescent staining of CN1 iPSC lines.

Rebuttal Figure 4. The immunofluorescent staining of CN1-derived NPC cells.

We also attempted to perform an additional replicate using a second guide RNA pair; however, despite screening over 1,000 monoclonal colonies over the past four months, we were unable to obtain a successful knockout line. Therefore, in this revision, we report results based only on the depletion achieved with a single guide RNA pair. We appreciate your understanding.

Recent studies highlighted potential biases in differential expression analysis tools, with some reporting that DESeq2 may yield a higher rate of false positives (Ge et al., 2021). To improve the robustness of DEG identification, we compared two widely used methods—DESeq2 and edgeR. Using these tools, we identified 546 upregulated and 862 downregulated genes by both tools (Supplementary Figure 26). Only 120 DEGs (8.52%) were discordant between the two analyses, indicating a high degree of consistency. Consistent with previous findings that DESeq2 tends to report more DEGs, our results support this observation. To ensure a more stringent and conservative analysis, we proceeded with the edgeR-derived gene set for downstream analyses.

Supplementary Figure 26. RNA-seq analysis of CN1-derived neural progenitor cells. (d) Comparison of raw and filtered differentially expressed genes (DEGs) identified by DESeq2 and edgeR. Venn diagrams show that while DESeq2 identified more DEGs than edgeR in both upregulated and downregulated categories, most DEGs were identified by both tools, demonstrating high consistency between two results.

You suggested that the variability in expression levels, particularly among TFs, should be carefully considered. Given that many TFs are lowly expressed and may exhibit elevated expression variability between replicates, you recommended examining the average expression level of the DEGs compared to other genes to ensure that their levels are typical and to minimize the concern about spurious DEG results. We added the following analysis to filter out the lowexpressed DEGs.

Supplementary Figure 26. RNA-seq analysis of CN1-derived neural progenitor cells. Permutation test on the mean transcript per million (TPM) of the raw upregulated (e) and downregulated DEGs (f) shows the low expression level of identified DEGs. Permutation test on the mean TPM of the filtered upregulated (g) and downregulated DEGs (h) indicates that there is no clear evidence of low expression for the filtered DEGs.

We performed a permutation test on the mean TPM of the up-regulated and down-regulated DEGs, respectively (sample size = 10,000). The results indicated that the mean expression levels of the identified up-regulated ($p = 0$) and down-regulated ($p = 0$) genes were significantly lower than those of the randomly sampled genes. Subsequently, we filtered out DEGs with TPM values in the lowest 50% of the entire gene expression set ($\text{TPM} > 1$), narrowing the DEG set to 99 upregulated and 178 downregulated DEGs. After filtering, we repeated the permutation test, and the results showed no evidence that the DEGs had lower expression levels than the randomly sampled genes ($p = 0.997$ for up-regulated DEGs and $p = 0.807$ for down-regulated DEGs) (supplementary figure 26).

These results support the reviewers' concerns. Thus, in the main text, we reported all GO analysis and genes after removing genes with $\text{TPM} < 1$. Meanwhile, the results of the GO analysis based on the filtered DEGs (Figure 5e-f) remain highly similar to those of the raw DEG set (Supplementary Figure 26i-j).

Figure 5. GO enrichment analysis of filtered upregulated genes (e) and downregulated genes (f) reveals that upregulated genes are associated with the pattern specification process, while downregulated genes are linked to pattern specification process and neuronal processes, such as forebrain development and axon guidance.

Supplementary Figure 26. RNA-seq analysis of CN1-derived neural progenitor cells. GO enrichment analysis of raw upregulated genes (h) and downregulated genes (i) reveals that upregulated genes are associated with the pattern specification process, while downregulated genes are linked to neuronal development and organization process. The results before filtering and after filtering are highly consistent.

All above results support the robustness of our experimental design and data analysis, reinforcing the validity of our claims. All analyses have been included in the main Figure 5 and Supplementary Figure 26, with further details provided in the Methods section (lines 536-580).

Another important validation is investigating the role of multi-mapping reads in the RNA-seq analyses. How were multi-mapping reads handled and is it possible that in the WT control the deleted region transcripts are stealing reads from other genomic locations that are then mapped to these locations when the deleted transcript are removed from the alignment target?

Thank you for raising this important point. However, we would like to clarify a few aspects of our analysis.

In our RNA-seq alignment, we did not remove transcripts from the deleted region when mapping reads from the depletion groups. Instead, we used the same reference genome and gene annotation file for both conditions (WT and depletion). This approach ensures consistency in read mapping across samples and prevents the so-called “stealing” phenomenon, where reads might otherwise be misassigned to alternative loci due to reference differences.

In addition, we identified SUNKs (single unique k-mers, k=21) within the exons of all genes located at the fusion sites. These SUNKs serve as evidence for unique read mapping. With the exception of *MIR1302-3*, all other genes contain SUNKs within their exons, supporting the reliable and unambiguous alignment of reads to these genes.

Rebuttal Table 1. Number of SUNKs in five genes

Gene	Number of SUNK
DDX11L2	320
FAM138B	10
MIR1302-3	0
PGM5P4	129
WASH2P	126

Regardless, it would be helpful to think about 5/10 DEGs validating by RT-PCR from a statistical perspective: is this an expected validation rate for false positives or indicative of many DEGs being true positives?

Our previous analysis of the fusion site KO was not sufficiently rigorous. In this revision, we have redesigned the KO experiments as detailed above. Following the suggestions from you and the other reviewers, we implemented an improved experimental design using three independent monoclonal lines, as described above. The results from these biological replicates show consistent clustering, supporting the robustness of our findings. Therefore, the current data provide evidence that depletion of the fusion site can affect gene expression.

6) The DEG results would be more impactful if the mechanism of differential expression and the potential consequences were explored a bit more deeply. Is transcription (or lack of transcription) of the (pseudo)genes in the SDs involved or not? Were any of the DEGs in other copies of SD_fusion_A, B or C on other chromosomes? Does removing the fusion region lead to generalized stress response? These are indeed important and interesting questions that we are also keen to explore. However, addressing them would require additional experiments, such as Hi-C and other molecular assays, to investigate the underlying mechanisms driving gene expression changes. These work would likely take 1-2 years to complete and are therefore beyond the scope of the current work.

Our current study supports the conclusion that depletion of the fusion site affects gene expression in NPCs. However, the molecular mechanisms underlying this observation remain unknown. To clarify this limitation, we have added the following sentence in the *Limitations of the Study* section (Line 445-448):

“While our study demonstrates that depletion of the fusion site leads to gene expression changes in NPC cells, the underlying molecular mechanisms—such as potential alterations in 3D genome structure or functional roles of genes within the fusion site—remain to be elucidated.”

7) Given the estimated timing of the fusion event and high degree of ILS, what role do you think that the emergence of human chr2 played in speciation?

The fusion of chr2 in modern humans and their ancestors likely contributed to reproductive isolation from NHP due to the concept of stasipatric speciation proposed by White and others. Furthermore, our DEG analysis suggests that this fusion event may have impacted gene regulatory networks. Intriguingly, this speciation event appears to coincide with ILS, prompting us to propose a hypothesis in which diverse ancestral polymorphic subtelomeric structures (ILS) may have contributed to the speciation process.

We envision the following scenario: Within the ancestral human–*Pan* population, which was genetically diverse and large in size, a chr2 fusion occurred in individuals carrying specific subtelomeric structures. Individuals harboring the fused chromosome likely experienced reduced reproductive compatibility with others in the population. Over time, this reproductive isolation mediated by a chromosomal incompatibility may have contributed to the emergence of a new lineage—modern humans’ ancestors—with ILS shaping the dynamics of early divergence.

In summary, our study provides evidence that ILS (large ancestral polymorphisms) is a major evolutionary force shaping subtelomeric structure formation, challenging previous hypotheses. In addition, our KO analysis points to a potential functional role of the chr2 fusion event. Although the KO results suggest that the fusion site may confer regulatory or adaptive advantages, we currently lack sufficient molecular mechanisms to support this interpretation. Therefore, we avoid overstating this speculative aspect in the main manuscript.

Reviewer #2:

In this manuscript, the authors report an extensive analysis of the structure and evolution of the fusion site on human chromosome 2. It has been known for many years that the largest karyotypic difference between humans and our great ape relatives is the unique fusion of two ancestral chromosomes to form modern human chromosome 2. The fusion site and a decayed centromere were mapped and described years ago. But due to the complex and highly repetitive structure of the fusion site on human chromosome 2, and the equally complex and repetitive nature of the relevant regions in the chimpanzee and gorilla chromosomes, it has been impossible to resolve in any detail the structure of the fusion site in humans, or the homologous structures in the apes.

Thank you for your time in carefully reviewing our manuscript.

Jiang et al. report here the use of the recently produced T2T genome assemblies for human, chimpanzee, bonobo, gorilla, two orangutan species and rhesus macaque to investigate the content and evolution of the fusion site. The manuscript describes in impressive detail and laudable clarity the complex structures of the relevant regions in the African apes and humans. The text provides a thorough explanation of the relevant telomeric and subtelomeric regions from the various species, as well as defining and characterizing the multiple copies of segmental duplications both within each single species genome and across species. The figures present useful diagrams of the relationships among the various chromosomal regions and outline the inferred evolutionary steps that led to both the differentiation of chromosome 2a and 2b content across the great apes and the evolutionary origin of the structure of the fusion region in humans. I find the descriptions of the regions to be thorough and sufficient. The documentation of the various structural elements is sufficient to provide the reader with an appreciation for the complexity and the dynamics of the relevant chromosomal segments. The functional analysis (i.e. comparing how gene transcription in HEK293T cells differs depending on presence or depletion of the fusion site) is interesting and significant as it illustrates the potential wider effect of the chromosomal fusion event. Overall this is a valuable contribution to understanding of both the evolution of the human genome and the mechanisms and genomic details of karyotypic change more generally.

We appreciate your positive comments.

The comparisons across species indicate that there is no simple evolutionary scenario that can explain the pattern of shared and differentiated chromosomal structures across the radiation of humans, chimpanzees, gorillas and orangutans. Rather, the authors argue that incomplete lineage sorting provides the only sufficient explanation for why the structure of the human fusion region is not a simple derivative of either the chimpanzee/bonobo structures or the gorilla structures. The authors invoke incomplete lineage sorting (ILS) to explain why the present-day distributions across species of SD_fusion_A, SD_fusion_B and SD_fusion_C cannot be represented as a simple evolutionary tree but rather imply different histories for different chromosomal segments. In my opinion, it is possible that ILS is the explanation for this complexity, but I do not believe it is the only possible explanation. While ILS might be a sufficient mechanism, there are now many published examples of introgression and hybridization between recently diverged primate and other mammalian species. Introgression from one species into a closely related species is (a) more common than was assumed years ago, (b) a sufficient explanation for the distinct histories of the various relevant chromosomal segments and (c) a process distinct from ILS.

We completely agree that ILS is not the only mechanism that could contribute to this process, but we would like to emphasize that ILS is indeed involved in the chromosome 2 fusion event.

To distinguish between introgression and ILS in the SD_fusion_A/B/C regions, we used mcmctree program in the software PAML (Yang et al., 2007; Reis et al., 2011) to estimate the divergence time. Specifically, we randomly selected one hundred 100 kbp genomic regions from human chromosome 2 and aligned them with orthologous regions in bonobo, chimpanzee, gorilla, Sumatran orangutan, and macaque. Of these, 93 regions were uniquely and successfully aligned and displayed a topology consistent with the species tree, and were retained for further analysis. To eliminate potential biases introduced by software algorithms (previously, we used BEAST2 for estimating the split time), we estimated the divergence time of the SD_fusion_A/B/C regions using mcmctree. Our analysis showed that the SD_fusion_A/B/C regions coalesced significantly earlier than other 93 randomly selected genomic regions. These results support our conclusion that ILS plays a role in the formation of the fusion site region.

Rebuttal Figure 5. Divergence time comparison between the species tree and SD_fusion_A/B/C supports the involvement of ILS in the formation of the fusion site region. Coalescent trees were constructed using *mcmctree* for SD_fusion_A (a), SD_fusion_B (b), and SD_fusion_C (c). (d) Distribution of Human–Pan and Human–Gorilla divergence times across the 93 control regions, with the Human–Pan divergence times of SD_fusion_A/B/C highlighted.

As you suggested, we also discussed other factors probably involved in chr2 fusion in the limitation of this study section at line 429-434:

“Speciation is not an instantaneous process—it typically requires extended periods of population divergence and is often more complex than expected. Our study demonstrates that SDs have undergone lineage-specific sorting in extant apes, including humans, and may have played a key role in facilitating the chr2 fusion in the (ancestral) human lineage. However, the exact timing of when this fusion became nearly fixed in the (ancestral) human population remains unknown, but our results suggest it is ancient and that it occurred early in African ape speciation >5 mya.

Our study supports the idea that ILS plays a crucial role in fusion site formation based on current genomic data from living apes. However, the large population size of the common ancestor of great apes complicates the evolutionary scenario. For example, recent studies have shown that ghost introgression occurred during great ape speciation. Therefore, we have sufficient reason to guess that ILS is not the only evolutionary force involved in this fusion formation, and other forces (e.g., ancestral introgression) could also be associated with the fusion. Importantly, our study is the first to propose that ILS is involved in chromosome 2 fusion and may have facilitated great ape speciation and the origins of humans.”

Jiang et al. found that there are multiple degenerate haplotypes present in the human population for the centromere that became non-functional after the chromosomal fusion. It is quite possible that at various times during the initial differentiation of the human, Pan and gorilla lineages that each incipient lineage/species had several haplotypes segregating for chromosomes 2a, 2b and the fusion. Gene flow from one incipient lineage into another seems to me to be just as plausible an explanation for the complex contradictory histories of the various chromosomal sub-regions. Furthermore, there is compelling evidence for additional African great ape lineages that went extinct, the so-called ghost lineages (e.g. Pawar et al PMID37500909 and Kuhlwilm et al. PMID31036897). The authors should either explain why it is not possible that during the early evolutionary differentiation of the extant lineages (periods when there were increasing genetic differences but still opportunities for inter-lineage gene flow) there could not have been introgression of distinct haplotypes from one lineage into another, involving either two of the lineages that survived to the present or one of those and a now extinct ghost lineage. It seems to me that the difference in ILS proportions proximal and distal to the fusion site provides some evidence that it was not just ILS that generated the complex pattern the authors reveal.

We completely agree that the evolutionary scenario is much more complicated than what we have uncovered in great apes. There are likely more ancient great ape species (extinct) and 'ghost' introgression events in great ape lineages (Kuhlwilm et al., 2019; Pawar et al., 2023). However, testing for ghost introgression in these complex regions is challenging, as it requires more population data to accurately estimate the allele frequencies of genotypes in these repetitive regions. Since we do not yet have these data, we are unable to conduct such experiments. Nevertheless, we fully agree with your point regarding the possible role of ghost introgression and a more complex evolutionary scenario. To clarify, we have added the following discussion in Line 429-434:

“Speciation is not an instantaneous process—it typically requires extended periods of population divergence and is often more complex than expected. Our study demonstrates that SDs have undergone lineage-specific sorting in extant apes, including humans, and may have played a key role in facilitating the chr2 fusion in the (ancestral) human lineage. However, the exact timing of when this fusion became nearly fixed in the (ancestral) human population remains unknown, but our results suggest it is ancient and that it occurred early in African ape speciation >5 mya.

Our study supports the idea that ILS plays a crucial role in fusion site formation based on current genomic data from living apes. However, the large population size of the common ancestor of great apes complicates the evolutionary scenario. For example, recent studies have shown that ghost introgression

occurred during great ape speciation. Therefore, we have sufficient reason to suppose that ILS is not the only evolutionary force involved in this fusion formation, and other forces (e.g., ancestral introgression) could also be associated with the fusion. Importantly, our study is the first to propose that ILS is involved in chromosome 2 fusion and may have facilitated great ape speciation and the origins of humans.”

In Line 323 the authors speculate that the fusion event had the effect of "...creating a stasipatric speciation barrier." This is also debatable. If the authors are implying an immediate barrier to interbreeding arose between the population in which the fusion occurred and the other closely related lineages, this is not reasonable. The fusion event would (most likely) have occurred only one time and heterozygotes for the ancestral condition and the fusion must have been fertile for many generations while the fused chromosome increased in frequency. This presumed fertility of heterozygotes means an immediate barrier to inter-population breeding is very improbable. The authors will have to provide a much stronger argument if they wish to infer a direct connection between the fusion event and "speciation."

There seems to be a misunderstanding regarding the use of the term 'sympatric'. We did not mean immediate speciation or a complete reproductive barrier between individuals with fused chromosome 2 and those without it. We hypothesize that these two groups could likely mate, but with reduced fitness, as you suggested.

To clarify, we have revised our sentence as follows in Line 358-362:

“Given fossil evidence that *Australopithecus* existed around 2-4 mya, *Paranthropus* around 1-3 mya, and the earliest *Homo* fossils around 2-3 mya, we speculate that this fusion event probably did not arise in the genus *Homo* but rather occurred in ancestral great ape populations that would give rise to humans potentially by creating a possibility of reduced interbreeding between the two karyotypes in ancestral populations, ultimately leading to speciation.”

Lesser concerns: There are a number of minor issues in the manuscript that should be addressed.

Thank you for your other important comments and please see the point-to-point response below.

Lines 117-118: I believe this should read "...while another occurred after the Pan-gorilla split."

We revised the text as you suggested in Line 129-131:

“One pericentric inversion occurred in the common ancestor of African great apes after divergence from orangutans (chr2b) while another occurred after the *Pan*-gorilla split (chr2a)”

Line 148 and after: The authors suggest that the date of the divergence of the gorilla lineage from the human-Pan lineage was about 7.4 mya. My reading of the literature suggests that the date should be older than that. Both Shao et al. (Science 380: 913, 2023) and Dos Reis et al. (Syst. Biol. 67: 594, 2018) put it at least 9.9 mya. Other papers also suggest a date older than 7.4 mya. The authors may have strong justification for the younger date. But if they do, they should cite the supporting prior studies.

We refer to Mao et al. (2024) for node calibration (Supplementary Table 16). The date of the divergence of the gorilla lineage from the human-Pan lineage was about 7 mya in this paper. In our previous study, we proposed that discrepancies in divergence time estimates may be attributed to incomplete lineage sorting (ILS) during the speciation of African or great apes. And we cite this paper in Line 203.

Based on these, we used BEAST with the HKY model incorporating gamma site, calibrated yule, and relaxed log-normal clock models to infer the split time. Node ages were estimated using lognormal priors (nodes settings see Supplementary Table 17). We conducted three independent runs for each tree and the results were consistent. Effective sample sizes exceeded 200 for all parameters in all runs.

Supplementary Table 16. Node ages estimated from 19 independent runs (ILS sites excluded) in the paper Mao et al., 2024.

Nodes	Mean*	95% CI*
H-M	28.32	26.31-30.34
H-O	14.05	13.11-15
H-G	7.03	6.47-7.58
H-P	5.00	4.55-5.45
H-B	1.45	1.32-1.58
H-C	1.45	1.32-1.58

H: human; M: macaque; O: orangutan; G: gorilla; P: Pan; B: bonobo; C: chimpanzee * The mean and 95% values are estimated based on the 19 independent runs.

Supplementary Table 17. Node ages used in time-calibration tree reconstruction.

SD_fusion_A		
	M	S
H-M	25	0.05
H-O	14	0.15
H-B	1.5	0.15
H-C	1.5	0.15
SD_fusion_B		
	M	S
H-M	25	0.05
H-O	14	0.15
H-B	1.5	0.15
H-C	1.5	0.15
SD_fusion_C		
	M	S
H-M	25	0.05

M: mean; S: standard deviation

Lines 321-322: I think the authors intend to say: "...we speculate that this fusion event did NOT arise in the genus *Homo*...."

You are correct. We revised it in Line 359-362:

"we speculate that this fusion event did not arise in the genus *Homo* but rather occurred in ancestral great ape populations that would give rise to humans potentially by creating a possibility of reduced interbreeding between the two karyotypes in ancestral populations, ultimately leading to speciation".

Minor issues:

Line 58: Reference #7 is listed in the reference list as Painter and Stone, not Wilson and Painter.

We revised these in the revised manuscript Line 63-64:

"In 1935, Painter and Stone first observed that chromosome fusion could lead to speciation in flies."

Line 191: Given how the field today is emphasizing telomere-to-telomere (T2T) assemblies, it might be better to drop that use of the T2T abbreviation for a different concept. I do not think that abbreviation is used again in the manuscript.

We agree with you and we revised the text as (Line 210-212):

"Previous investigations identified inverted telomeric sequences (TTAGGG/CCCTAA) at the fusion site, suggesting a telomere-to-telomere fusion"

Reviewer #3:

In this paper the authors use the completed great ape sequences (to be published in another paper) to reconstruct the genomic events that led to the fusion of human chromosome 2. This is a complicated (and wonderfully nerdy :-)) endeavor given the complexity and details of the reconstruction of the duplicated, inverted, rearranged and twisted SD regions. It is also as the authors rightly say " arguably the most significant karyotypic difference between humans and NHPs" and hence in my opinion absolutely justified, interesting and relevant to get the region and its evolutionary history straight. Together with the follow-up on the transcriptional effects of the deletion, I think that this is a very good and adequate paper for Cell Genomics.

As a disclaimer, I cannot fully judge the correctness of all details of the complex reconstructions, but given the expertise of the authors and the presented results, this makes a very reasonable impression. This said, I have two major issues that should be improved before publication Thank you for your time in carefully reviewing our manuscript.

A) Clarity of text

While I find the Introduction very clear and understandable, I think parts of the paper including the discussion can profit from a more clear and understandable writing. It is beyond the scope of my review to suggest rewritings, but maybe if the person who wrote the introduction could also improve the other parts, the manuscript and hence authors, readers, journal would profit quite some bit. Here are a few minor points I came across:

Thank you for your suggestions. In line with similar feedback from Reviewer #1, we have substantially revised the writing throughout the manuscript. Specifically, we added opening sentences to each Results subsection to improve clarity and added transitional statements in the Discussion to enhance logical flow.

L47-48: "Most conspicuously, one of these SDs shares homology to a hypomethylated SD spacer sequence present in hundreds of copies in the subterminal heterochromatin of chimpanzees and bonobos" It's not clear to me in what way this is conspicuous, i.e. which mechanism/process/evolution/hypothesis is informed by this finding.

We want to emphasize the relationship between the fusion site and *Pan* subterminal regions. As per your suggestion, we revised this sentence as in line 49-52:

“Most interestingly, one of these SDs shares homology to a hypomethylated SD spacer sequence present in hundreds of copies in the subterminal heterochromatin of chimpanzees and bonobos, but is completely absent subtelomerically in both humans and orangutans.”

L51-52: change to "...alters the expression of genes in HEK cells" as I think it is important that this is in HEK cells

We agree that the cellular context is important when interpreting gene expression changes. HEK cells, due to their origin and genetically heterogeneous nature, may undermine the robustness of our conclusions. To improve the confidence of our findings, we redesigned the entire experiment using CN1 neuronal progenitor cells (NPCs) cells.

The CN1 cell line is a human induced pluripotent stem cell (iPSC) line described by Yang et al., 2023 (PMID: 37452091). We performed immunofluorescent staining for stem cell and NPC marker genes in the cell lines used in this study. As shown below, all cell lines exhibited clear expression of the corresponding marker genes (Rebuttal Figure 3 and 4).

Rebuttal Figure 3. The immunofluorescent staining of CN1 iPSC lines.

Rebuttal Figure 4. The immunofluorescent staining of CN1-derived NPC cells.

We selected three independent monoclonal iPSC-CN1 cell lines and performed CRISPR-Cas9 editing using a single guide RNA pair (L-sg1&R-sg). For the control group, we added scrambled sgRNAs. After validating successful depletion, we differentiated both knockout (KO, n = 3) and control (CTRL, n = 3) cell lines into NPCs and performed RNA-seq on each line. RNA expression clustering revealed that KO samples grouped together, and control samples clustered separately, indicating that the knockout is the primary factor contributing to the differences between the control and KO groups (Supplementary Figure 26). This revised experimental design using iPSC-CN1 offers improved genetic clarity and robustness over our initial experiments in HEK293 cells.

Supplementary Figure 26. RNA-seq analysis of CN1-derived neural progenitor cells. (b) Principal component analysis (PCA) of RNA-seq replicates using the top 500 most expressed genes shows strong replication within each condition. (c) The Euclidean distance matrix illustrates the consistency within groups and large differences between groups.

We also attempted to perform an additional replicate using a second guide RNA pair; however, despite screening over 1,000 monoclonal colonies over the past four months, we were unable to obtain a successful knockout line. Therefore, in this revision, we report results based only on the depletion achieved with a single guide RNA pair. We appreciate your understanding.

We revised this sentence as below:

“CRISPR/Cas9-mediated depletion of the fusion site in human cell lines significantly alters the expression of genes in human neural progenitor cells (NPCs)”

L55: what is meant by "with ILS in speciation"? A bit unclear sentence, especially for the last one of the abstract..

We revised it as the following (Line 57-60):

“Overall, this study provides a detailed understanding of the genomic structure, evolutionary history, and functional implications of the human chromosome 2 fusion, offering the first insights into how complex regions subject to ILS may contribute to speciation.”

L74 difference (singular) not differences?

We revised it as “difference”.

L154: I think when ILS first appears, one should quickly remind reads what that is and maybe give a reference where they can read it up, as I am not sure that all of the readers interested in the article will be familiar with this phenomenon

We added a description of ILS at its mention in this study.

Line 196-200:

“Incomplete lineage sorting (ILS) refers to the random segregation of ancestral alleles into descendant lineages, a process that typically arises when the ancestral population size is large.

ILS results in gene tree discordance and deep coalescence. Previous studies have speculated that ILS may facilitate speciation, although direct evidence and underlying mechanisms remain to be fully explored.”

L 322: should be: "... we speculate that this fusion event did not arise in the genus *Homo*..."

You are correct and we revised it in Line 359-362:

“we speculate that this fusion event did not arise in the genus *Homo* but rather occurred in ancestral great ape populations that would give rise to humans potentially by creating a possibility of reduced interbreeding between the two karyotypes in ancestral populations, ultimately leading to speciation.”

B) Fusion site depletion

I think the depletion of the fusion site and the following RNA-seq is really a highlight of the paper as it goes beyond the pure sequence reconstruction and the most important biological result from the paper. But I think the experiment is not described sufficiently and also not replicated sufficiently:

Based on your valuable suggestions, we redesigned the fusion site depletion experiments, and these additional efforts have substantially improved the quality of our work.

In this revision, we performed CRISPR-Cas9 editing using a single guide RNA pair (L-sg1&Rsg). For the control group, we added scrambled sgRNAs. After validating successful depletion, we selected three

independent monoclonal iPSC-CN1 cell lines for both knockout (KO, $n = 3$) and control (CTRL, $n = 3$) cell lines and differentiated them into NPCs and performed RNA-seq on each line. RNA expression clustering revealed that KO samples grouped together, and control samples clustered separately, indicating that the knockout is the primary factor contributing to the differences between the control and KO groups (Supplementary Figure 26). This revised experimental design using iPSC-CN1 offers improved genetic clarity and robustness over our initial experiments in HEK293 cells.

Supplementary Figure 26. RNA-seq analysis of CN1-derived neural progenitor cells. (b) Principal component analysis (PCA) of RNA-seq replicates using the top 500 most expressed genes shows strong replication within each condition. (c) The Euclidean distance matrix illustrates the consistency within groups and large differences between groups.

We performed immunofluorescent staining for stem cell and NPC marker genes in the cell lines used in this study. As shown below, all cell lines exhibited clear expression of the corresponding marker genes (Rebuttal figure 3 and 4).

Rebuttal Figure 3. The immunofluorescent staining of CN1 iPSC lines.

Rebuttal Figure 4. The immunofluorescent staining of CN1-derived NPC cells.

We also attempted to perform an additional replicate using a second guide RNA pair; however, despite screening over 1,000 monoclonal colonies over the past four months, we were unable to obtain a successful knockout line. Therefore, in this revision, we report results based only on the depletion achieved with a single guide RNA pair. We appreciate your understanding.

Recent studies highlighted potential biases in differential expression analysis tools, with some reporting that DESeq2 may yield a higher rate of false positives (Ge et al., 2021). To improve the robustness of DEG identification, we compared two widely used methods—DESeq2 and edgeR. Using these tools, we identified 546 upregulated and 862 downregulated genes by both tools (Supplementary Figure 26). Only 120 DEGs (8.52%) were discordant between the two analyses, indicating a high degree of consistency. Consistent with previous findings that DESeq2 tends to report more DEGs, our results support this observation. To ensure a more stringent and conservative analysis, we proceeded with the edgeR-derived gene set for downstream analyses.

Supplementary Figure 26. RNA-seq analysis of CN1-derived neural progenitor cells. (d) Comparison of raw and filtered differentially expressed genes (DEGs) identified by DESeq2 and edgeR. Venn diagrams show that while DESeq2 identified more DEGs than edgeR in both upregulated and downregulated categories, most DEGs were identified by both tools, demonstrating high consistency between two results.

The reviewer#2 suggested that the variability in expression levels, particularly among TFs, should be carefully considered. Given that many TFs are lowly expressed and may exhibit elevated expression variability between replicates, you recommended examining the average expression level of the DEGs compared to other genes to ensure that their levels are typical and to minimize the concern about spurious DEG results. We added the following analysis to filter out the low-expressed DEGs.

Supplementary Figure 26. RNA-seq analysis of CN1-derived neural progenitor cells. Permutation test on the mean transcript per million (TPM) of the raw upregulated (e) and downregulated DEGs (f) shows the low expression level of identified DEGs. Permutation test on the mean TPM of the filtered upregulated (g) and downregulated DEGs (h) indicates that there is no clear evidence of low expression for the filtered DEGs.

We performed a permutation test on the mean TPM of the up-regulated and down-regulated DEGs, respectively (sample size = 10,000). The results indicated that the mean expression levels of the identified up-regulated ($p = 0$) and down-regulated ($p = 0$) genes were significantly lower than those of the randomly sampled genes. Subsequently, we filtered out DEGs with TPM values in the lowest 50% of the entire gene expression set ($\text{TPM} > 1$), narrowing the DEG set to 99 upregulated and 178 downregulated DEGs. After filtering, we repeated the permutation test, and the results showed no evidence that the DEGs had lower expression levels than the randomly sampled genes ($p = 0.997$ for up-regulated DEGs and $p = 0.807$ for down-regulated DEGs) (Supplementary Figure 26).

These results support the reviewer#2s' concerns. Thus, in the main text, we reported all GO analysis and genes after removing genes with $\text{TPM} < 1$. Meanwhile, the results of the GO analysis based on the filtered DEGs (Figure 5e-f) remain highly similar to those of the raw DEG set (Supplementary Figure 26i-j).

Figure 5. GO enrichment analysis of filtered upregulated genes (e) and downregulated genes (f) reveals that upregulated genes are associated with the pattern specification process, while downregulated genes are linked to pattern specification process and neuronal processes, such as forebrain development and axon guidance.

Supplementary Figure 26. RNA-seq analysis of CN1-derived neural progenitor cells. GO enrichment analysis of raw upregulated genes (i) and downregulated genes (j) reveals that upregulated genes are associated with the pattern specification process, while downregulated genes are linked to neuronal development and organization process. The results before filtering and after filtering are highly consistent.

All above results support the robustness of our experimental design and data analysis, reinforcing the validity of our claims. All analyses have been included in the main Figure 5 and Supplementary Figure 26, with further details provided in the Methods section (lines 536-580).

All above additional work probably can address your concerns about the fusion site KO analysis.

Detailed descriptions of these experiments and analyses for your additional comments are provided in the point-by-point responses below.

1) A better description of what was actually done is needed. The method section is unclear how exactly cells were selected and it is very unclear how RNA-seq was done (it is just mentioned that total RNA was reverse transcribed and normalized to GAPDH, but this refers to the QPCR, I guess!).

Thank you for your advice. In general, for the experiment using CN1 cell lines, we generated fusion site knockout (KO) iPSC lines using a single guide RNA pair (L-sg1&R-sg). For the control group, we added scrambled sgRNAs. After validating successful depletion, we selected three independent monoclonal iPSC-CN1 cell lines for both knockout (KO, n = 3) and control (CTRL, n = 3) cell lines and differentiated them into NPCs and performed RNA-seq on each line.

We revised the Method as the following (Line 536-580):

“**Fusion site KO experiments.** The design of optimal sgRNA pairs to target sites was performed using the online CRISPR design tool (<http://crispor.gi.ucsc.edu/>). The complementary oligonucleotide pairs of L-sgRNA #1/#2 and R-sgRNA were annealed at 95°C for 5 min, with ramp-down to 25°C to generate the double-stranded DNA (dsDNA) fragment, before ligation into BbsI-digested PX459 (plasmid #48139, Addgene).

Wild-type induced pluripotent stem cells (CN1) or mutant induced pluripotent stem cells were maintained in mTeSRTM Plus (Stemcell). Cells were dissociated into single-cell suspensions using TrypLE (Gibco) and electroporated using the Lonza Nucleofector 4D X Unit (program CA137) and the P3 Primary Cell Nucleofection Kit (V4XP-3024). The following conditions were used: $5\text{-}6 \times 10^5$ cells/ml, 10 μg sgRNA (two sgRNAs mixed at a 1:1 ratio). The control groups were electroporated with scramble sgRNAs. After electroporation, iPSCs were resuspended in mTeSRTM Plus supplemented with 10 μM ROCK inhibitor Y27632 and plated onto 12-well culture plates within 24 h. At 48 h post-electroporation, the culture medium was replaced with fresh mTeSRTM Plus containing puromycin (1 $\mu\text{g}/\text{ml}$) for 48h of selection. Cells were subsequently maintained in standard mTeSRTM Plus medium until colony formation.

We selected three independent monoclonal iPSC-CN1 cell lines for the knockout (KO, $n = 3$) and control (CTRL, $n = 3$) condition. For isolation of KO or control monoclonal clones, selected colonies were dissociated into single-cell suspensions. Individual cells were manually transferred using a mouth pipette under a stereomicroscope and seeded into 96-well plates. When a single cell grows into a clonal colony, we extract the genomic DNA from the cells with the TIANamp genomic DNA Kit (DP304-02, Tiangen) according to the manufacturer’s instructions. Genome deletion was detected by PCR-amplification of gDNA using a primer pair flanking the deletion. The primers used for PCR screening are listed in Supplementary Table 8. Genomic PCR products were detected by agarose gel electrophoresis and Sanger sequencing (Shanghai, Tsingke).

iPSCs Differentiation into Neural Progenitor Cells. Neural Progenitor Cells (NPCs) were generated from control iPSCs or KO iPSCs using the STEMdiff™ Neural System (Stemcell). Briefly, Cells were harvested using TrypLE and resuspension in STEMdiff™ Neural Induction Medium + SMADi + 10 μM Y-27632 as single cells. Add cell suspension (2×10^6 cells/well) to a single well of the matrix-coated 6-well plate and then perform a daily full-medium change with warm (37°C) medium until cultures are ready to be passaged. At passage 3, NPCs were expanded in STEMdiff™ Neural Progenitor Medium.

RNA Preparation and RNA-seq data analysis

Total RNA was isolated from each replicate of the KO cell lines and the control cell lines using Hieff NGS Ultima Dual-mode mRNA Library Prep Kit, following the manufacturer’s instructions. We utilized RNA-seq pipeline from the nf-core community with default parameters to align and quantify the RNA-seq data. Differential expression analysis was conducted using the R package DESeq2 and edgeR. We identified differentially expressed genes (DEGs) with following criteria: false discovery rate (FDR) < 0.05 and a log2 fold change > 2. Because edgeR and DESeq2 produced similar results, we used the results by edgeR to do the further analysis. We performed Gene Ontology (GO) enrichment analysis using clusterProfiler.”

To be honest, the description does not suggest a lot of experience with RNA-seq analyses as for example the amount of produced Gbp is mentioned in the main text, but what matters and is usually reported is the number of reads.

We revised it and added the number of reads in Line 315-319:

"For each condition (wild type and depletion), we selected three independent monoclonal cells as biological repeats, we then conducted RNA-seq, generating 100.3, 106.2 and 104.6 million reads for the control cell lines, and 137.8, 130.3 and 133.1 million reads for the fusion site deletion cell lines (Supplementary Figure 26a-c and Supplementary Table 10)."

It is also very uncommon to use two statistical programs (limma and edgeR) and take select genes that are significant in both.

Recent studies highlighted potential biases in differential expression analysis tools, with some reporting that DESeq2 may yield a higher rate of false positives (Ge et al., 2021). To improve the robustness of DEG identification, we compared two widely used methods—DESeq2 and edgeR. Using these tools, we identified 546 upregulated and 862 downregulated genes by both tools. Only 120 DEGs (8.52%) were discordant between the two analyses, indicating a high degree of consistency (Supplementary Figure 26). Consistent with previous findings that DESeq2 tends to report more DEGs, our results support this observation. To ensure a more stringent and conservative analysis, we proceeded with the edgeR-derived gene set for downstream analyses.

Supplementary Figure 26. RNA-seq analysis of CN1-derived neural progenitor cells.

Comparison of raw differentially expressed genes (DEGs) identified by DESeq2 and edgeR. Venn diagrams show that while DESeq2 identified more DEGs than edgeR in both upregulated and downregulated categories, most DEGs were identified by both tools, demonstrating high consistency between two results.

Furthermore, QPCR from the same clones does not hurt, but is not thought to be actually necessary as QPCR is less sensitive than properly performed RNA-seq. However, what is of great importance is the type of replicates and it is not clear from the text how lines of wt and modified HEK cells were chosen and hence it is unclear what was replicated.

In the new experiment using CN1 cell lines, we selected three independent monoclonal iPSCCN1 cell lines for the knockout (L-sg1&R-sg KO, n = 3) and control (CTRL, n = 3) condition (three biological replicates for each condition). For further details, please refer to the information above. As you suggested, we do not perform QPCR validations in the revision.

2) The central claim is that deletion causes expression changes in HEK cells. To be valid, this needs to be significant, i.e. needs to be found in independent replications of this experiment. As the description of what was done is not clear (see above), I assume that 3 wells from one singlecell isolate of sg2, one from sg1 and culture from the wildtype ancestors was done.

As we responded above, in this revision, we generated a control and two KO CN1 lines (Lsg1&R-sg and L-sg2& R-sg), and isolated three independent monoclonal clones for each condition. While we successfully established L-sg1&R-sg KO monoclonal clones, we failed to obtain monoclonal clones for the L-sg2&R-sg condition, despite attempts in the past four months (we repeated this several time and picked over thousands of monoclonal clones, but still failed to the KO monoclonal clones). All clones were then induced into NPCs under identical conditions and sent for bulk RNA sequencing.

These updates have been incorporated into the Methods section (line 536-580), with the revised results presented in the main Figure 5 and Supplementary Figure 26.

What is relevant is that the heterogeneity in HEK cells (or any cell culture) in combination with single cell cloning can lead to substantial differences in expression that are not related to the genetic manipulation performed but due to genetic and epigenetic properties of the single clone.

We conducted the experiment using the CN1 cell line, which exhibits clearer and more stable genetic and epigenetic properties compared to HEK cells (Yang et al., 2023).

We performed immunofluorescent staining for stem cell and NPC marker genes in the cell lines used in this study. As shown below, all cell lines exhibited clear expression of the corresponding marker genes (Rebuttal Figure 3 and 4).

Rebuttal Figure 3. The immunofluorescent staining of CN1 iPSC lines.

Rebuttal Figure 4. The immunofluorescent staining of CN1-derived NPC cells.

Notably, the relatively small overlap between the two lines sg1 and sg2 suggests this to be the case (although difficult to judge given the limited description of the experiments). I think to really nail the point that the fusion causes expression changes, one would need to make RNA-seq of at least six single-cell cloned HEK lines that carry the deletion (either sg1 or sg2, assuming that they have identical effects) and six that did not carry it (i.e. WT clones that underwent the same procedure e.g. with scrambled sgRNAs or at a minimum derived from a single clone of the same population). Note that I find the proper design of the experiment the crucial point and not the result. The analysis is as interesting to be published also if one finds no DEGs

We first thank you for recognizing the value of our evolutionary analysis and for your constructive criticisms regarding the KO analysis, which were also raised by the other two reviewers. As detailed above, we redesigned the experiments in accordance with your and the other reviewers' suggestions. The updated results support the robustness of our experimental design and data analysis, reinforcing the validity of our conclusions. All revised analyses have been incorporated into the main Figure 5 and Supplementary Figure 26, with additional details provided in the Methods section (lines 536-580).

3) HEK cells are actually not kidney epithelial cells, but probably adrenal cells of neuronal origin (not sure, but I think this is the proper reference: <https://pmc.ncbi.nlm.nih.gov/articles/PMC4166678/>). Not that this would devalue the experiments per se, but I think it is important to mention.

As you suggested, we used IPSC-CN1 and NPC-CN1 cell lines rather than HEK in the revised study.

4) HEK cells are also known to have massive genomic rearrangements; Given the topic of the paper, I think it would be important to address how many copies of the fusion site are actually present in HEK cells

To avoid the variable copy number of fusion sites in HEK cells, as you suggested, we used IPSC-CN1 and NPC-CN1 cell lines rather than HEK in the revised study.

Referees' reports, second round of review

Reviewer #1:

My suggestions have largely been addressed. Continuing to edit the text in Results to enhance clarity around the interpretation of each result would enhance the impact.

Reviewer #2:

I think this paper is very strong and that this revision is an improvement over the first version. The authors have addressed my issue concerning the possibility of introgression and also dealt with the other reviewer critiques satisfactorily. However, I believe that the authors are over-interpreting their important and interesting results in one quite significant way. I fully agree with the authors that the results indicate that incomplete lineage sorting (ILS) played a substantive role in producing the between-

species distribution of genomic features that these authors document among the human, chimpanzee, bonobo, gorilla and orangutan lineages. The data show that SD_fusion_A, _B and _C display a pattern among the extant species that is best explained as the result of ILS in combination with other population genetic processes. However, I read various statements by the authors to be arguing that this ILS was actually a cause, a "driver" in their words, of the speciation event that separated ancestral humans from the ancestors of extant genus Pan. I am convinced that ILS helps to explain the presence or absence of various genomic segments among these various lineages, so I agree with the authors that as the various lineages diverged and became independent species, that ILS is part of the explanation for the pattern of genomic diversity observed when comparing human chromosome 2 with chimpanzee, bonobo and gorilla chromosomes 2a and chromosome 2b.

But I interpret the statement in lines 359-362 to be arguing that that the fusion event was possibly a cause of the divergence of species. Lines 420-421 make this argument unambiguously. I respectfully suggest that there is no evidence presented here for this inference. The fact that only one surviving lineage among the multiple surviving lineages generated from the ancestral African great ape population now carries this chromosome fusion does not implicate that chromosomal feature in the process of reproductive isolation that led to speciation. To argue that this fusion event was the driver of a speciation event requires that the authors show reason to conclude that this chromosomal fusion created a barrier to interbreeding, and to simultaneously show how that fused chromosome could increase in frequency in a population, if it was sufficient to block inter-breeding between populations. We can presume that the fusion occurred in a single individual and was passed on to offspring, present as heterozygous in all carriers for several generations prior to reaching an allele frequency high enough to generate homozygotes. That increase in allele frequency in the population depends on the reproductive success of heterozygotes, and thus the authors must explain how heterozygotes enjoyed breeding success for many generations, but that inter-population gene flow through the continued breeding of heterozygous hybrids was blocked. In their Response to Reviewers, the authors argue that heterozygous carriers of the fusion would have reduced although still effective reproductive success. But that is the paradox. I am asking how the fertility of heterozygotes is high enough to increase the frequency of the fused chromosome in the original population where it occurred, but so low that it "drives" speciation between divergent populations. This is not addressed in the manuscript.

Furthermore, I note that some of the authors of this paper recently published another excellent paper reporting that more than 39% of the human genome exhibits ILS in relation to the African apes (Yoo et al. 2025; PMID 40205052). This chromosome 2 fusion is just one locus among a large number across the human genome that exhibit ILS. Why is this one the "driver" of speciation?

I suggest that the authors have not fully appreciated the relationship between ILS and the divergence of evolutionary lineages. ILS is not a cause of differentiation but is one result when a population with high levels of genetic variation divides into three or more descendent lineages over a short time period. The authors mistakenly attribute the idea that "ILS may facilitate speciation" (line 199) to Rivas-Gonzalez et al 2023 (citation #50 in the Yang et al. manuscript). The paper by Rivas-Gonzalez et al. describes the use of ILS data to investigate various aspects of the ancient evolutionary genomics of primates, including dating of speciation events and patterns of positive and negative selection. But that 2023 paper does not suggest that ILS "facilitates" or can be a cause of speciation events itself. Rivas-Gonzales and colleagues show that ILS can be used to learn about the history of evolutionary divergences, but do not suggest it is a factor contributing to the origin of reproductive isolation and species divergence. Yang et al. also cite a paper on plant speciation, but it is widely recognized that speciation, the origin of

reproductive barriers, hybridization and other related evolutionary processes operate very differently in plants than in primates, so I do not consider a citation to speciation mechanisms in plants to be relevant to the issue here.

I would suggest that it is fine for the authors to explicitly raise the possibility that this chromosome fusion event may have contributed at some level to evolutionary divergence of human ancestors from the ancestors of the genus *Pan*. The data Yang et al. have on changes in gene expression resulting from depletion of the fusion region argue for possible phenotypic effects relevant to hominin evolution. But the authors should make clear to readers that there is no direct evidence for this idea that the fusion caused the speciation and that it is really just speculation about one possible impact of the fusion. To state that "...the SD insertions in the human fusion site, moreover, may have driven the speciation between humans and NHPs through ILS..." (lines 420-421) is unsupported speculation at best.

This is a minor issue, but in several places, the authors use the phrase "nonhuman great apes" or "nonhuman African great apes" (lines 117, 124, 180). I think this is redundant. Just "great apes" or "African great apes" should be fine.

Reviewer #3:

The authors made the requested clarifications and improvements, especially in the context of the RNA-seq analyses; I think the paper can be published

Authors' response to the first round of review**Reviewer #2:**

I think this paper is very strong and that this revision is an improvement over the first version. The authors have addressed my issue concerning the possibility of introgression and also dealt with the other reviewer critiques satisfactorily. However, I believe that the authors are over-interpreting their important and interesting results in one quite significant way. I fully agree with the authors that the results indicate that incomplete lineage sorting (ILS) played a substantive role in producing the between-species distribution of genomic features that these authors document among the human, chimpanzee, bonobo, gorilla and orangutan lineages. The data show that SD_fusion_A, _B and _C display a pattern among the extant species that is best explained as the result of ILS in combination with other population genetic processes. However, I read various statements by the authors to be arguing that this ILS was actually a cause, a "driver" in their words, of the speciation event that separated ancestral humans from the ancestors of extant genus *Pan*. I am convinced that ILS helps to explain the presence or absence of

various genomic segments among these various lineages, so I agree with the authors that as the various lineages diverged and became independent species, that ILS is part of the explanation for the pattern of genomic diversity observed when comparing human chromosome 2 with chimpanzee, bonobo and gorilla chromosomes 2a and chromosome 2b.

But I interpret the statement in lines 359-362 to be arguing that that the fusion event was possibly a cause of the divergence of species. Lines 420-421 make this argument unambiguously. I respectfully suggest that there is no evidence presented here for this inference. The fact that only one surviving lineage among the multiple surviving lineages generated from the ancestral African great ape population now carries this chromosome fusion does not implicate that chromosomal feature in the process of reproductive isolation that led to speciation. To argue that this fusion event was the driver of a speciation event requires that the authors show reason to conclude that this chromosomal fusion created a barrier to interbreeding, and to simultaneously show how that fused chromosome could increase in frequency in a population, if it was sufficient to block inter-breeding between populations. We can presume that the fusion occurred in a single individual and was passed on to offspring, present as heterozygous in all carriers for several generations prior to reaching an allele frequency high enough to generate homozygotes. That increase in allele frequency in the population depends on the reproductive success of heterozygotes, and thus the authors must explain how heterozygotes enjoyed breeding success for many generations, but that inter-population gene flow through the continued breeding of heterozygous hybrids was blocked. In their Response to Reviewers, the authors argue that heterozygous carriers of the fusion would have reduced although still effective reproductive success. But that is the paradox. I am asking how the fertility of heterozygotes is high enough to increase the frequency of the fused chromosome in the original population where it occurred, but so low that it "drives" speciation between divergent populations. This is not addressed in the manuscript.

Furthermore, I note that some of the authors of this paper recently published another excellent paper reporting that more than 39% of the human genome exhibits ILS in relation to the African apes (Yoo et al. 2025; PMID 40205052). This chromosome 2 fusion is just one locus among a large number across the human genome that exhibit ILS. Why is this one the "driver" of speciation? I suggest that the authors have not fully appreciated the relationship between ILS and the divergence of evolutionary lineages. ILS is not a cause of differentiation but is one result when a population with high levels of genetic variation divides into three or more descendent lineages over a short time period. The authors mistakenly attribute the idea that "ILS may facilitate speciation" (line 199) to Rivas-Gonzalez et al 2023 (citation #50 in the Yang et al. manuscript). The paper by Rivas-Gonzalez et al. describes the use of ILS data to

investigate various aspects of the ancient evolutionary genomics of primates, including dating of speciation events and patterns of positive and negative selection. But that 2023 paper does not suggest that ILS "facilitates" or can be a cause of speciation events itself. Rivas-Gonzales and colleagues show that ILS can be used to learn about the history of evolutionary divergences, but do not suggest it is a factor contributing to the origin of reproductive isolation and species divergence. Yang et al. also cite a paper on plant speciation, but it is widely recognized that speciation, the origin of reproductive barriers, hybridization and other related evolutionary processes operate very differently in plants than in primates, so I do not consider a citation to speciation mechanisms in plants to be relevant to the issue here. I would suggest that it is fine for the authors to explicitly raise the possibility that this chromosome fusion event may have contributed at some level to evolutionary divergence of human ancestors from the ancestors of the genus *Pan*. The data Yang et al. have on changes in gene expression resulting from depletion of the fusion region argue for possible phenotypic effects relevant to hominin evolution. But the authors should make clear to readers that there is no direct evidence for this idea that the fusion caused the speciation and that it is really just speculation about one possible impact of the fusion. To state that "...the SD insertions in the human fusion site, moreover, may have driven the speciation between humans and NHPs through ILS..." (lines 420-421) is unsupported speculation at best.

Thank you very much for reviewing our manuscript. We also sincerely appreciate your valuable comments. In some parts of the text, we acknowledge that there is no direct evidence supporting the statements, as they are hypotheses. To avoid potentially misleading readers, we have revised the writing as follows:

Line 359–362:

Original:

"We speculate that this fusion event probably did not arise in the genus *Homo* but rather occurred in ancestral great ape populations that would give rise to humans potentially by creating a possibility of reduced interbreeding between the two karyotypes in ancestral populations, ultimately leading to speciation⁶⁰."

Revised:

"We speculate that this fusion event probably did not arise in the genus *Homo*, but rather occurred in ancestral great ape populations."

Line 420–421:**Original:**

“The divergent fates of the subtelomeric repetitive regions of these ancestral chromosomes, e.g., pCht expansion and SD spacer insertions in the Pan lineage, or the SD insertions in the human fusion site, moreover, may have driven the speciation between humans and NHPs through ILS (Supplementary Figure 30).”

Revised:

“The divergent fates of the subtelomeric repetitive regions of these ancestral chromosomes— e.g., pCht expansion and SD spacer insertions in the Pan lineage, or the SD insertions in the human fusion site— may have contributed to divergence between humans and NHPs, potentially involving ILS (Supplementary Figure 30).”

Line 199:**Original:**

“Previous studies have speculated that ILS may facilitate speciation, although direct evidence and underlying mechanisms remain to be fully explored^{50, 51}.”

Revised:

“Previous studies have speculated that ILS may be involved in evolutionary divergence, although direct evidence and the underlying mechanisms remain to be fully explored^{50, 51}.”

This is a minor issue, but in several places, the authors use the phrase "nonhuman great apes" or "nonhuman African great apes" (lines 117, 124, 180). I think this is redundant. Just "great apes" or "African great apes" should be fine.

We specifically refer to nonhuman great apes and nonhuman African great apes in these sentences, as humans also belong to the great ape family. We have chosen to use these terms for clarity. Thank you for your understanding.